# Optimization of the Micellar-Based In Situ Gelling Systems Posaconazole with Quality by Design (QbD) Approach and Characterization by In Vitro Studies

**DOI:** 10.3390/pharmaceutics14030526

**Published:** 2022-02-27

**Authors:** Meltem Ezgi Durgun, Burcu Mesut, Mayram Hacıoğlu, Sevgi Güngör, Yıldız Özsoy

**Affiliations:** 1Department of Pharmaceutical Technology, Faculty of Pharmacy, Istanbul University, Istanbul 34126, Turkey; mezgi.kilic@istanbul.edu.tr (M.E.D.); bmesut@istanbul.edu.tr (B.M.); sgungor@istanbul.edu.tr (S.G.); 2Department of Pharmaceutical Microbiology, Faculty of Pharmacy, Istanbul University, Istanbul 34126, Turkey; mayramtuysuz@gmail.com

**Keywords:** Posaconazole, micelle, in situ gelling systems, rheological behavior, ocular drug delivery system, poloxamer 40, poloxamer 188, TPGS

## Abstract

Background: Fungal ocular infections can cause serious consequences, despite their low incidence. It has been reported that Posaconazole (PSC) is used in the treatment of fungal infections in different ocular tissues by diluting the oral suspension, and successful results were obtained despite low ocular permeation. Therefore, we optimized PSC-loaded ocular micelles and demonstrated that the permeation/penetration of PSC in ocular tissues was enhanced. Methods: The micellar-based in situ gels based on the QbD approach to increase the ocular bioavailability of PSC were developed. Different ratios of Poloxamer 407 and Poloxamer 188 were chosen as CMAs. T_sol/gel_, gelling capacity and rheological behavior were chosen as CQA parameters. The data were evaluated by Minitab 18, and the formulations were optimized with the QbD approach. The in vitro release study, ocular toxicity, and anti-fungal activity of the optimized formulation were performed. Results: Optimized in situ gel shows viscoelastic property and becomes gel form at physiological temperatures even when diluted with the tear film. In addition, it has been shown that the formulation had high anti-fungal activity and did not have any ocular toxicity. Conclusions: In our previous studies, PSC-loaded ocular micelles were developed and optimized for the first time in the literature. With this study, the in situ gels of PSC for ocular application were developed and optimized for the first time. The optimized micellar-based in situ gel is a promising drug delivery system that may increase the ocular permeation and bioavailability of PSC.

## 1. Introduction

Due to the complex structure of the eye and its natural barriers, the bioavailability of drugs used in the treatment of ocular diseases is quite low. Especially, about 80% of a topically applied drug is rapidly eliminated by the tear film and nasolacrimal drainage [1]. Conversely, it is not possible for the drug remaining without elimination to reach the target area by bypassing the other barriers of the eye. To overcome these limitations, the development of ocular drug delivery systems is a challenging issue that attracts the attention of pharmaceutical technologists [2,3,4].

Micelles are nanocarriers that have been shown to be effective as an ocular drug delivery system and are available as a commercial product. Micelles, consisting of a hydrophobic core and a hydrophilic shell, are formed by the self-aggregation of amphiphilic copolymers or surfactants above the critical micelle concentration. Their hydrophobic core enables the transport of lipophilic drugs to hydrophilic tissues [1,5].

In situ gelling systems are drug carrier systems that are in solution before being applied to the body but become gel form after application, depending on factors such as the temperature and pH of the environment. Due to the bioadhesive properties of the polymers used in its preparation, the contact time with the tissue increases, and they show extended release [6,7,8]. In addition, since they are in liquid form before application and their viscosity increases after application, the application difficulties of viscous drugs are not seen in in situ gelling systems. In situ gels produced with polymers such as carbomers, hydroxypropyl methyl celluloses, and poloxamers, which have been used in drug preparation for years, are reliable dosage forms in terms of excipients. They are often preferred, especially in ocular dosage form since they do not impair the visual function of the patient due to the transparent structures formed when they are in gel form [9,10].

PSC is the triazole group member anti-fungal agent with the widest spectrum [11]. Oral suspension, solution for infusion, and tablet form are commercially available in the market [12]. Although none of these commercial products are suitable for ocular application, Noxafil^®^ oral suspension is used when diluted to treat severe ocular fungal infections with a high potential to blind [13,14]. For this reason, we developed a PSC-loaded micellar drug delivery system for ocular application in our previous studies [15,16]. The most important reasons for choosing micelles as a carrier system in our study:Because of the high lipophilic character of PSC, micelles are a suitable carrier system for the eye that contain hydrophilic and lipophilic tissues together [1];The particle size of the micelles can be adjusted in accordance with the pore size of the ocular tissues [1];The presence of different ocular commercial products with micellar structures in the market [17,18,19,20];Micelles can be produced with reliable copolymers such as TPGS that have received GRAS approval [21].

In recent studies, drug delivery systems have been used in combination [22]. Due to the combined use, the synergetic effect of the advantages of different carrier systems can be achieved. Thus, the maximum bioavailability profile can be reached at a minimum dose. This study turned the PSC-loaded ocular micelles, which we had previously optimized, into an in situ gelling system using appropriate polymers. Our previous studies improved the water solubility, drug release profile, ocular permeation, penetration, and safety of PSC via micelles. However, this optimized formulation is in the form of a solution and still carries the risk of elimination, even if it is not as much as a conventional dosage form when applied topically. We planned to rule out this elimination risk by improving the optimized micellar formulation’s in situ gelling systems. We think that both the optimized micellar formulation and the micellar-based in situ gelling systems we will develop are an important alternative to Noxafil^®^ oral suspension, which is used off-label to treat ocular fungal infections in the treatment of ocular fungal infections in the clinic. For this reason, we used the quality by design (QbD) method while developing in situ gels to minimize the margin of error and to allow for commercialization in the future.

The development of a drug is basically about identifying critical properties of patient needs. At this point, the quality-by-design approach enables drug design to be made with parameters for the quality of the final product for the patient’s needs [23]. With the QbD approach, first, the quality target product profile (QTPP) is determined, the parameters of the product quality (CQA) are determined, and then the manufacturing parameters that will affect the quality are determined [24]. This study aimed to develop a PSC in situ gel formulation to be used in the treatment of ocular fungal disease. The polymer type (Poloxamer 407 and Poloxamer 188) and ratios used were determined as critical material properties (CMAs), and their effects on critical quality attributes (CQAs) were investigated. Characterization, rheologic behavior analysis, mechanical properties, in vitro drug release, anti-fungal activity, and HET-CAM toxicity analysis of the QbD-optimized PSC-loaded micellar-based in situ gelling system were performed.

## 2. Materials and Methods

PSC was kindly gifted from Deva Pharmaceutical Company (Küçükçekmece, Istanbul, Turkey), which was purchased from MSN Laboratories (Whitefields, Kondapur, Hyderabad, India). TPGS, Poloxamer 407, Poloxamer 188, and Carbopol 980 were kindly gifted from BASF (Ludwigshafen, Germany). Sodium lauryl sulfate, sodium chloride, sodium bicarbonate, calcium chloride dihydrate, sodium phosphate dibasic, sodium phosphate monobasic, methanol (high-performance liquid chromatography (HPLC) grade, and acetonitrile (HPLC) grade were purchased from Merck (Darmstadt, Germany) and were used as received. Hydroxypropyl methylcellulose (50M, 60M, 75HD100), methylcellulose (MC), and sodium carboxymethylcellulose (NaCMC) were kindly gifted from Ashland Chemicals (Wilmington, DE, USA). All other chemicals were of analytical grade. The ultrapure water was supplied from Millipore Milli-Q ultrapure water system (Billerica, MA, USA), and isotonic 0.9% NaCl solution was purchased from Polifarma (Küçükçekmece, Istanbul, Turkey).

### 2.1. Preparation and Characterization of PSC-Loaded TPGS Micelles

The PSC-loaded TPGS micellar formulation used in the study was selected among the formulations we prepared and optimized in our previous studies. The formulation coded NM-51 was used in these studies, which gave the best results. PSC as 250 µg per mL and TPGS as 15 mg per mL were weighed and dissolved in methanol and acetonitrile, respectively. These two solutions were mixed at room temperature. The mixture was evaporated with a rotary evaporator (BÜCHI Labortechnik GmbH, Essen, Germany) under vacuum at 55 °C. The thin film layer was hydrated with 0.9% NaCl isotonic solution and filtered through a 0.45 μm PTFE membrane filter (Macherey-Nagel GmbH & Co KG, Duren, Germany). In the characterization studies, it was seen that the particle size, PDI value, and zeta potential of the micelles were compatible with our previous study [15].

### 2.2. Pre-Formulation Studies of Micellar-Based In Situ Gelling Systems

Thermosensitive polymers of different structures were used to obtain PSC-loaded micellar-based ocular in situ gels. In pre-formulation studies, poloxamer 407, poloxamer 188, different types of HPMC (50M, 60M, 75HD100), MC, and NaCMC were added to the micellar formulation alone or combined in different proportions. Studies have also been carried out with Carbopol 980, a pH-sensitive polymer. Thus, the effect of different polymer types on the formulation was investigated. Appropriate amounts of polymers were weighed and added onto the PSC-loaded micelles mixed with a magnetic stirrer to obtain PSC-loaded micellar-based in situ gelling systems. All formulations were mixed overnight at +4–8 °C [25]. The pH of the in situ gels was measured and was adjusted to 7.4, which is appropriate for the ocular application, with 0.1N NaOH solution.

### 2.3. Preparation of PSC-Loaded Micellar-Based In situ Gelling Systems

According to the results obtained from the pre-formulation studies, it was decided to use poloxamer 407 and poloxamer 188 single or in combination as thermosensitive gelling agents in the preparation of in situ gels. To perform optimization studies with QbD, a total of 15 formulations were studied with both polymers at different concentrations (15–20% *w*/*v*). To obtain PSC-loaded micellar-based in situ gelling systems, appropriate amounts of poloxamer 407 and/or 188 were weighed and added onto the PSC-loaded micelles mixed with a magnetic stirrer. All formulations were mixed overnight at +4–8 °C [25]. The pH of the in situ gels was measured and was adjusted to 7.4, which is appropriate for the ocular application, with 0.1N NaOH solution.

### 2.4. Characterization of In Situ Gelling Systems

#### 2.4.1. Clarity, Gelling Capacity, and Drug Content

The produced in situ gels were examined in terms of clarity, gelling capacity, and drug content (DC). To determine gelling capacity, 100 µL of in situ gel was added into 2 mL of simulated tear fluid (STF) at 37 °C [26,27]. The gelling capacity of the formulations was analyzed over time and evaluated in five different categories. The gelling capacity categories are given in Table 1. DC of in situ gels was quantified by high-pressure liquid chromatography (HPLC) (LC 20AT, Shimadzu, Kyoto, Japan) as described previously [15] and was calculated according to Equation (1).
DC% = A × 100/B (1)

A: The amount of PSC loaded into the drug delivery systems (μg).

B: The amount of PSC used for the preparation of the drug delivery systems (μg).

#### 2.4.2. Rheology

The rheological properties of the formulations were performed using a controlled stress rheometer (Haake Rheometer I, Thermo Fisher Scientific Inc., Essen, Germany) with a parallel steel cone-plate geometry (35°TiL and 0.052 mm gap distance). All analyses were performed at 35 °C to mimic ocular physiological temperature. The oscillation study was also studied at 5 and 60 °C, the start and end temperatures of the T_sol/gel_ analysis.

Continuous shear (flow) analysis was performed to obtain a curve on in flow mode over shear rates ranging from 0 to 2000 s^−1^. The measurement was made for 150 s [28]. The shear stress (Pa), the consistency index-k (Pa.s), the rate of shear (s^−1^), and the flow behavior index (n) were calculated using RheoWin 4.87.0006 (Haake^®^) software.

Thixotropy analysis was performed to obtain two curves were obtained in flow mode over shear rates ranging from 0 to 2000 s^−1^ and 2000 to 0 s^−1^. During the analysis, ascent and descent curves were determined within 150 s of each [28]. When the system reached the value of 2000 s^−1^, it was kept for 60 s at 2000 s^−1^. The hysteresis area was calculated using RheoWin 4.87.0006 (Haake^®^) software.

The viscoelastic properties of the in situ gels were evaluated in the oscillating mode. The linear viscoelastic region (LVR) of each sample at 5, 35, and 60 °C was determined. For this purpose, amplitude sweep analysis and frequency sweep analysis were performed between 0.01–100 Pa and 0.1–10.0 Hz, respectively [28,29]. Storage modulus (G′), loss modulus (G″), dynamic viscosity (η′), and loss tangent (tan δ) were obtained using RheoWin 4.87.0006 (Haake^®^) software.

T_sol/gel_ determination was performed in oscillating mode at a temperature range of 5–60 °C. A controlled heating rate (5 °C/min) was used in the study. Temperature measurements were made in 10 steps, and the system was kept for 5 min at each step. The analysis was performed at a frequency of 1.0 Hz and shear stress of 1.0 Pa, according to the LVR values obtained in the oscillation studies [28,29]. G′, G″, η′ and tan δ were calculated using RheoWin 4.87.0006 (Haake^®^) software. T_sol/gel_ value was accepted as the temperature at which G′ and G″ were equal (G′ and G″ crossover).

### 2.5. In Situ Gel Formulation Development Based on the Quality by Design and Optimization

The first step of QbD is the determination of QTPP. QTPP summarizes the product characteristics, and the QTPP details of the study are given in Table 2.

In order to understand the interaction between the variables correctly, it is necessary to create a correct design area [30]. To see this interaction, a three-level full factorial design was applied. Conversely, to see only one surfactant’s effects on the formulations, the formulations were produced containing only 1 surfactant.

Gelation properties are the risky parameters for in situ gel formulations, and the used polymer type and ratio directly affect these properties [31]. In this study, Poloxamer 188 and Poloxamer 407 and different ratios of these polymers were chosen as critical material attributes (Table 3).

According to the QTPP, the critical CQA parameters are determined. In addition, risky critical material attributes (CMA) and critical process parameters (CPP) are decided. In our study, 15 formulations were prepared, and gelling capacity, T_sol/gel_ temperature, Log consistency index, and drug loading capacity were chosen as critical quality attributes (CQAs).

Fifteen formulations were prepared, and Gelling Capacity, Tsol/gel temperature, Log consistency index, and Drug loading capacity were chosen as Critical Quality Attributes (CQAs).

Formulations were prepared, and CQAs were tested. While conducting QbD studies, Tsol/gel temperature, gelling capacity, drug content and log consistency index were used as CQAs. The obtained data were transferred to Minitab 18 program (Minitab, Chicago, IL, USA), and the relations between input and output variables were evaluated. The regression coefficient (R^2^), adjusted R^2^, and predicted R^2^ values were calculated from the program. *p* values and Pareto charts were also used to understand the interaction between inputs and outputs. The optimized formulation was determined by Minitab 18, and the CQAs responses for the formulation were analyzed.

### 2.6. Characterization of Optimized In Situ Gel

#### 2.6.1. Clarity, Gelling Capacity, and Drug Content

Optimized in situ gel was analyzed in terms of clarity, gelling capacity, and drug content as described in the title of Section 2.4.1 of this article.

#### 2.6.2. Rheology

The rheological properties of the optimized formulation were performed as de-scribed in the Section 2.4.2. using a controlled stress rheometer (Haake Rheometer I Thermo Fisher Scientific Inc., Essen, Germany) with a parallel steel cone-plate geometry (35°TiL and 0.052 mm gap distance). However, the rheology studies of the optimized in situ gel were carried out at 3 temperatures, different from the Section 2.4.2. All analyses were performed at 5, 25, and 35 °C to mimic storage, room, and ocular physiological temperature, respectively. In situ gel will be diluted with tear film after applied to the eye. For this reason, oscillation studies and T_sol/gel_ studies were also studied with in situ gel diluted with STF at 35 °C different from the Section 2.4.2. Dilution was made in situ gel:STF(50:7) when the volume of a drop and the tear volume were calculated [32].

### 2.7. Texture Profile Analysis (TPA)

The mechanical properties (cohesion, adhesion, hardness, and compressibility) of the optimized formulation were determined at 5 ± 1, 25 ± 1, and 35 ± 1 °C using the Brookfield CT3 tissue analyzer. A ball-shaped analytical probe with a diameter of 25.4 mm was used for 5 ± 1 and 25 ± 1 °C analyses (TA43). Then, 10 g of the formulation was placed in a 25 mL beaker. The probe was immersed twice at a depth of 5 mm in the sample, with a velocity of 2.0 mm/sec and an initiating force of 0.01 N, within a time interval of 15 s. The measurements were conducted in triplicate for both temperatures. Since optimized in situ gel (IS-OPT) was in gel form at 35 ± 1 °C, a cylindrical analytical probe with a diameter of 12.7 mm was used in the analysis (TA5). Then, 10 g of the formulation was placed in a 25 mL beaker. The probe was immersed twice in the sample at a depth of 10 mm, with a velocity of 2.0 mm/sec and an initiating force of 0.01N, in a time interval of 15 s. The measurements were conducted in triplicate. Using the obtained power–time curve, the hardness, compressibility, adhesiveness, and cohesiveness of the gels were calculated. TexturePro CT V1.6 Build software (AMETEK Brookfield, Middleborough, MA, USA) was used for data collection and calculation [33].

The hardness of gels is the force required to deform the gel. In the texture analysis, the highest power value obtained in the first period of penetration of the probe into the sample gives the hardness value (Newton, N) required for the deformation of the gels. The compressibility of the gels is the work required to deform the gel. In the texture analysis, the area under the first curve obtained during the first period of penetration of the probe into the sample gives the compressibility value (Newton × mm) of the gel. The adhesiveness of the gels is the work required to overcome the tensile forces between the surface of the gel and the surface of the probe. In texture analysis, the area under the second curve obtained during the first period of penetration of the probe into the sample after the first period of immersion in the sample gives the adhesion value (Newton × mm). The cohesiveness of the gels is calculated by dividing the area under the fourth curve obtained in the second immersion period of the probe into the sample in the texture analysis, by the area under the curve obtained in the first immersion period (AUC 1-2), and it has no units [34].

### 2.8. In Vitro Release Study

In vitro release studies were performed using a membraneless model [35,36]. Then, 2 mL in situ gel formulation was placed in test tubes with a diameter of 1.6 cm and a height of 12 cm with a capacity of 15 mL. The solution was gelled by preheating the test tubes at 35 °C for 10 min. At 35 °C, 2 mL of STF was added onto the gelling formulations as the release medium. The test tubes were placed upright on the horizontal shaker. The samples were shaken at 100 rpm on a horizontal shaker under a controlled temperature of 35 ± 0.5 °C. Every 20 min, the test tubes were removed from the shaker and all the release medium was taken and replaced with another 2 mL of the release medium. This process was continued until the gels in the test tubes were completely dispersed. Instead of the sample taken, 100 µL of the release medium was added to the vials. Samples were then diluted to 1 mL with the mobile phase and analyzed in HPLC as previously described [15]. The cumulative PSC amount released from the gels was calculated.

### 2.9. Anti-Fungal Activity Studies—Time-Kill Assay

The rate of killing the inoculums of *C. albicans* ATCC 90,028 was determined by incubating the test organism with agitation in RPMI-1640 (Sigma-Aldrich, St. Louis, MO, USA) supplemented with L glutamine and buffered with morpholinepropane-sulfonic acid (MOPS) (Sigma-Aldrich, St. Louis, MO, USA) containing diluted Noxafil^®^ oral suspension, in situ gelling system or micelles. Prior to time-kill evaluation, C. albicans was subcultured on Sabouraud Dextrose agar (SDA) (Difco, Sparks, MD, USA) plates, and following incubation for 24 h, colonies were suspended in RPMI-1640 medium and were added to each tube to give a final inoculum concentration of 1 × 10^6^ CFU/mL. One tube containing only fungal inoculum of 1 × 10^6^ CFU/mL was prepared as control. The tubes were incubated on a shaker water bath at 35 °C and at an agitation speed of 70 rpm. The samples were removed at predetermined time points (0, 1, 2, 3, 4, 5, 6, and 24 h) and diluted in sterile saline, if necessary. Then, 100 µL aliquots were placed onto SDA plates and incubated at 35 °C for 24–48 h. After incubation time, surviving colonies were counted (CFU/mL), and the time-kill curves were constructed by plotting the log10 colony counts against time. The lower limit of detection for time-kill assays was one log10 CFU/mL. Antimicrobial carry over effect was determined as formerly reported [37,38].

### 2.10. HET-CAM Toxicity Tests

The HET-CAM test is a toxicity study that is conducted with fertilized White Leghorn chicken eggs. It was developed as an alternative to the Draize Rabbit Eye Test to minimize the number of animals to be used in toxicity tests [39]. The HET-CAM study was carried out in accordance with the ICCVAM guidelines [40]. Fresh fertile White Leghorn chicken eggs (not older than seven days) weighing 50–60 g were used in toxicity tests. Eggs were incubated at 37.8 ± 0.3 °C and 58 ± 2% relative humidity in an automatic rotary incubator for nine days. At the end of the 9th day, the eggs were checked, and the dead or defective eggs were discarded. The air cell sections of each egg were marked. Then, the eggshells were cut from the marked part with a rotating dental saw. The cut pieces were carefully removed to avoid damaging the inner membrane.

In toxicity tests, ultrapure water, 0.9% NaCl isotonic solution, 0.1 N NaOH solution, PSC-loaded micellar-based in situ gel and its placebo form were used. Since the HET-CAM toxicity test of diluted Noxafil^®^ oral suspension, PSC-loaded micelles and their placebo form were performed in our previous study; it was not repeated. Then, 0.9% NaCl isotonic solution and 0.1 N NaOH solution were used as negative and positive controls, respectively. Then, 0.3 mL of the formulations and control solutions were applied to the vascular chorioallantoic membranes (CAM) of eggs. CAM reactions were observed over a period of 300 s, and the time at which each of the indicated endpoints (lysis, bleeding, or coagulation) occurred was recorded [39]. The toxicity of the formulation was scored according to the time of the endpoints (Table 4). Each sample (*n* = 6) was scored separately. Finally, the mean score for that group was calculated. The toxicity of each in situ gel and placebo was then assessed by score (Table 5).

### 2.11. Statistical Analysis

The results are presented as a mean ± standard deviation (SD) of at least three experiments. The statistical difference was performed using t test and one-way ANOVA, followed by Bonferroni multiple comparison tests (GraphPad Prism Software (8.0.1), La Jolla, CA, USA). *p* value < 0.05 was considered as the level of statistical significance.

## 3. Results

### 3.1. Pre-Formulation Studies of Micellar-Based In Situ Gelling Systems

Pre-formulation studies were performed using poloxamer 407, poloxamer 188, different types of HPMC (50M, 60M, 75HD100), MC, NaCMC, and Carbopol 980 in different proportions alone or combination [6,7,8,28]. The results obtained are summarized in Table 6. While designing the formulation ingredients, poloxamer 407 and poloxamer 188 were determined as the main thermosensitive gelling agents. Other excipients were preferred as auxiliary agents. During the experiments, auxiliary gelling agents in formulations that did not gel even when heated to 75 °C or could not obtain a clear image were eliminated. In the first studies, it was observed that the formulations produced with poloxamer 407, poloxamer 188, 50M HPMC, and 60M HPMC showed gelation at certain temperatures and were completely clear. It was decided to continue the studies with these excipients. However, in situ gels containing both types of HPMC showed aggregation over time when scaled-up. This aggregation was in the form of gelation, and it was independent of the storage conditions. The aggregation was seen at both +4 °C, and 25 °C. It was reversible, and the gels regained their solution form when mixed. However, this was considered a stability problem, and all in situ gels containing HPMC were eliminated in the pre-formulation study. For this reason, it was decided to produce formulations to be developed and optimized with QbD with poloxamer 407 and poloxamer 188.

### 3.2. Characterization of In Situ Gelling Systems

#### 3.2.1. Clarity, Gelling Capacity, and Drug Content

Clarity, gelling capacity, and drug contents of the PSC-loaded micellar-based in situ gels are given in Table 7. In addition, the contents of the formulations selected after the pre-formulation study and the final pH values after adjustment with 0.1 N NaOH are also included in the table.

#### 3.2.2. Rheology

The results of the rheology studies of PSC-loaded micellar-based in situ gels produced with different ratios of poloxamer 407 and/or poloxamer 188 with a controlled stress rheometer (Haake Rheometer I Thermo Fisher Scientific Inc., Essen, Germany) are given in Table 8.

### 3.3. In Situ Gel Formulation Development Based on the Quality by Design and Optimization

The R^2^, adjusted R^2,^ and predicted R^2^ values obtained for each CQAs parameter at the end of the evaluation made by the Minitab 18 program are given in Table 9 (the model equations are given as Appendix A). The Table (Table 10) of *p* values used to evaluate the significance of the models is given below. Interactions between inputs (CMAs) and outputs (CQAs) were investigated using Pareto charts and 2D contour graphs (Figure 1 and Figure 2). Optimized formulation was determined by Minitab 18, and software recommended Poloxamer 407 20% (*w*/*v*) and Poloxamer 188 0.404% (*w*/*v*) usage as an optimum formulation. Although there are no data for 0.404% (*w*/*v*) in formulation studies for Poloxamer 188, the program determined this concentration by evaluating formulations without Poloxamer 188.

### 3.4. Characterization of Optimized In Situ Gel

#### 3.4.1. Clarity, Gelling Capacity, and Drug Content

After the optimized in situ gel was prepared, its pH was adjusted to 7.4. The gel had a completely clear appearance. In the gelling capacity analysis performed in 2 mL of STF, it was determined that IS-OPT immediately gelled when dropped, and the formed gel retained its continuity for several hours (++ value). The drug content was found to be 90.97% (Table 7).

#### 3.4.2. Rheology

The results of the rheology studies of optimized in situ gelling system with a controlled stress rheometer (Haake Rheometer I Thermo Fisher Scientific Inc., Essen, Germany) are given in Table 11 and Figure 3. The T_sol/gel_ value of IS-OPT and T_sol/gel_ value after dilution with STF were found to be 31.56 ± 1.40 and 33.91 ± 3.30, respectively.

### 3.5. Texture Profile Analysis (TPA)

The mechanical properties of the formulations at storage temperature (5 °C), room temperature (25 °C), and ocular temperature (35 °C) were determined using a Brookfield tissue analyzer. With the help of the force–time curves obtained, the mechanical properties of the formulations such as hardness, compressibility, adhesiveness, and cohesiveness were determined (Table 12).

### 3.6. In Vitro Release Study

In vitro release studies were continued until the entire in situ gel was dissolved. At the end of the 3rd hour, the gels in all sample tubes had dissolved. The amount of PSC released from the in situ gel was calculated cumulatively. At the end of the 3rd hour, it was observed that a total of 71.6352% of the PSC in the gel systems was released (Figure 4). In vitro release graph is given in Figure 4. To determine the in vitro drug release model, the parameters in the study of Gouda et al. [41] were examined. It has been shown that the release of PSC from optimized micellar-based in situ gel follows the zero degree kinetic model (R^2^ = 0.9498).

### 3.7. Anti-Fungal Activity Studies—Time Kill Assay

The results were shown as time-kill curves in Figure 5. For comparison, the growth of a control of the *C. albicans* was also assessed. It was found that the control developed normally during the analysis. Time-kill analysis data demonstrated that all compounds showed anti-fungal activity as time dependent. IS-OPT and NM-51 decreased yeast counts by more than one log compared to control at 2 and 3 h, respectively. IS-OPT also showed more than one log reduction in *C. albicans* numbers at 24 h, compared with the control.

### 3.8. HET-CAM Toxicity Tests

Micellar formulations of TPGS and Poloxamer 407 have previously been prepared for many ocular applications. However, PSC-loaded ocular micelles were developed by our team for the first time. There are also no studies on the safety of ocular use of PSC recorded in the literature. For this reason, we performed the toxicity studies of the micelles we developed in our previous study and Noxafil^®^ oral suspension, which is used off-label in the clinical ocular area, using the HET-CAM method. Therefore, in this study, we only studied the toxicity of the optimized in situ gelling system and placebo. No signs of lysis, hemorrhage or coagulation were found in any of the samples at the end of 300 s in the studies. Images of the HET-CAM tests are given in Figure 6.

## 4. Discussion

In situ gels are in solution form during application. Depending on the environment’s temperature, pH, or ion balance conditions, they become gel in the application area. Thus, the residence time in the target tissue is increased. The bioavailability of drugs applied topically to the eye is low due to ocular barriers. In situ gelling systems are often preferred to increase bioavailability. The dilution of the drug, which gels after it is applied to the eye, with the tear film slows down, and the elimination effect of nasolacrimal drainage is reduced. Thus, the contact time and bioavailability of drugs with ocular tissues are increased. In this study, we developed thermosensitive in situ gels of PSC-loaded TPGS micelles that we had previously developed and optimized. While designing our formulas, we determined Poloxamer 407 and Poloxamer 188 as the main gelling agents.

Although Poloxamer 407 is a thermosensitive copolymer with the rheological properties expected from a gel, it does not have good mucoadhesive properties. For this reason, it may lose its gelling ability as soon as it is diluted with tear film after application [42]. To prevent this situation, the rate of poloxamer 407 used in the formulation can be increased. However, when used at 20–30% (*w*/*w*) ratios, the T_sol/gel_ value is below room temperature. This is also another undesirable situation. In this combined use, either poloxamer 188, which is thermosensitive itself, or mucoadhesive polymers such as different derivatives of methylcellulose and Carbopols are preferred. We tried both options in our pre-formulation studies.

Poloxamer 188 is used at a rate of ≥20% (*w*/*w*), the T_sol/gel_ value rises above 40 °C and cannot exhibit the rheological properties expected from a gel [10]. For this reason, it is common practice in many studies to use Poloxamer 407 in combination with poloxamer 188 at appropriate rates to bring the T_sol/gel_ value to the physiological range. We determined the rates of poloxamer 407 and 188 used in our study as 15–20% (*w*/*v*). Since our polymer usage ratio is determined as weight/volume, the T_sol/gel_ temperature values of formulations are compatible with previous studies using poloxamer 407 and 188 at similar rates [43]. Although the T_sol/gel_ value of an in situ gel to be applied ocular may differ in some sources, it is expected to be in the range of 30–35 °C. In our studies, the formulation containing a single 20% (*w*/*v*) poloxamer 407 gave results in this range.

In combining different mucoadhesive polymers with Poloxamer 407, another pillar of pre-formulation studies, we used different types of HPMC (50M, 60M, and 75HD100), MC, NaCMC, and Carbopol 980. It was observed that the formulations using 75HD100 HPMC, MC, NaCMC, and Carbopol 980 did not become gel at temperatures exceeding 75 °C, and/or did not have a clear appearance. The absence of gelation is thought to be due to the high T_sol/gel_ temperatures of the auxiliary gelling agents used [44,45]. Although Carbopol 980 is a pH-sensitive gelling agent, it is used in combination with Poloxamer 407 due to its excellent mucoadhesive properties. Thus, with the synergistic effect combining from both polymers, an in situ gel that is both thermosensitive and pH-sensitive is obtained. However, gelation was not observed at the physiological temperatures and pHs at the rates used. By increasing the Poloxamer 407 ratio, we could adjust the T_sol/gel_ temperature to the desired ranges. However, the fact that the obtained formulation was not clear was also a significant problem. Conversely, since the acidic nature of Carbopol can stimulate ocular tissues, we also avoided increasing the formulation rate [42]. For these reasons, we also eliminated Carbopol 980 in pre-formulation studies. Perhaps the most critical finding of the pre-formulation studies was that, despite the high T_sol/gel_ value, we observed that HPMC (50M and 60M) formulations that gelled when used with an appropriate amount of Poloxamer 407 precipitated during scale-up. We think that this situation is also caused by NaCl. The collapse, which cannot be seen clearly when studied at small scales, can be easily observed at large scales.

We conducted studies to evaluate the effect of Poloxamer 188 in combination with mucoadhesive polymers. The results obtained in these studies with Poloxamer 188, which was used at similar rates with Poloxamer 407, were not different from those obtained with Poloxamer 407. We did not expect any gelation when using only Poloxamer 188 with a higher T_sol/gel_ temperature than Poloxamer 407 in formulations. However, the fact that the formulations using 75HD100 HPMC, MC, NaCMC, and Carbopol 980 did not have a clear appearance, or the precipitation in the formulations containing 50M and 60M HPMC after scale-up showed that these problems were caused by other polymers, not Poloxamers.

According to the results obtained in the pre-formulation studies, we performed further studies with formulations in the IS1-IS15 range where Poloxamer 407 and Poloxamer 188 were used alone or in combination. All formulations in the range between IS1 and IS15 had a clear appearance, and drug content was found to be ≥90%.

One of the critical parameters for in situ gelling systems is the gelling temperature of the formulation. It is expected that the developed formulation will transition from solution to gel form at physiological temperatures. The expected optimum T_sol/gel_ temperature value for in situ gels to be applied ocular is between 30–35 °C [10]. The T_sol/gel_ value of all formulations developed in the IS1–IS15 range was determined. In the studies, it was seen that IS2, IS3, and IS13 showed gelation in the ocular physiological temperature range. T_sol/gel_ values of IS2, IS3, and IS13 were found to be 35.75, 30.36, and 35.51 °C, respectively. The data obtained are compatible with previous studies [10,29]. It was observed that the T_sol/gel_ value decreased as the Poloxamer 407 ratio in the formulations increased. In formulations containing single Poloxamer 188, gelation was not observed at physiological temperatures because the T_sol/gel_ value of Poloxamer 188 was high. In addition, the T_sol/gel_ value increased as the Poloxamer 188 ratio in the formulations increased. The changes in T_sol/gel_ value of formulations in the IS7-IS15 range, in which Poloxamer 407 and 188 were used in combination, were determined in parallel with the ratios of Poloxamer 407 and 188.

In situ gels are designed to switch from solution form to gel after application to the eye and increase contact time with ocular tissues. However, after application, they are eroded by tears and blinking action. For this reason, how long the formulations can maintain their integrity in the physiological environment is an important parameter that needs to be examined. Gelling capacity determination is an in vitro study performed with STF at ocular temperatures, mimicking the ocular environment. Thus, it gives preliminary information about the ability of the drug to preserve its integrity in ocular tissues. The gelation moment and complete dissolution of formulation added into STF are followed in the experiment. As soon as the formulation is completely dissolved, the experiment is terminated. In some studies, even if the formation of the gel is followed, it is recorded when the gel is completely dissolved [27]. In some studies, grouping is made according to the rate of gel formation and dissolution time [8]. Both gel formation rate and dissolution time are effective parameters in QbD studies. For this reason, we preferred the grouping method that examines both parameters. The studies determined that the gelling capacity was affected by the amount of both Poloxamer types used in the formulations. As the T_sol/gel_ temperature decreased as the amount of Poloxamer 407 increased, instant gelation was observed in the formulations we dropped into the STF in the gelling capacity studies. However, these gels formed quickly dissolved. This is due to the low gelling capacity of Poloxamer 407 when used alone less than 25% (*w*/*w*), as stated in the literature, and its rapid dilution with STF [32,42]. Since the T_sol/gel_ temperature was above the physiological temperature in formulations containing Poloxamer 188 alone, no gel formation was observed in the determination of gelation capacity. Similar results were obtained with T_sol/gel_ temperature in formulations where Poloxamer 407 and 188 were used in combination. IS13 with a T_sol/gel_ value of 35.51 °C gelled immediately as it was added to the STF. Another important point is that IS13 retained its gel structure for several hours, unlike IS2 and IS3. This result is compatible with the knowledge that the gelation capacity can be increased by using Poloxamer 407 in combination with Poloxamer 188, as stated in the literature [42].

For any formulation to be applied topically, its viscosity is significant. A drug with a low viscosity can flow before it becomes gel as it is applied, or it can be quickly eliminated with the tear film. Conversely, a drug with a high viscosity cannot be applied to the eye comfortably because it is not fluid [29]. It is important to produce a formulation with optimum viscosity to increase bioavailability in an organ with a small volume and high turnover, such as the eye. Conversely, the viscosity value of an in situ gel alone is not sufficient data to speculate about the behavior of that gel in vivo. Because viscosity studies are carried out at high speed, deformation occurs in the structure of the formulation. However, once the in situ gel is applied and spread over the eye surface, the mechanically affected parameter is the blink rate. For this reason, oscillation measurements with a low oscillation angle should be made to understand better the performance of an in situ gel in in vivo [46]. In the light of this information, the rheological properties of in situ gels were examined in detail in this study. The rheology properties of all formulations in the IS1–IS15 range were examined at the ocular temperature, 35 °C. In rheological examinations, it was determined that all formulations showed the Herschel–Bulkley flow characteristic, which is a non-Newtonian flow type.

Hysteresis areas were calculated in thixotropy analyses. The hysteresis area usually reveals the reversibility of the formations’ response to shear, showing two different profiles: thixotropic (positive hysteresis area) or rheopectic (negative hysteresis area) [47,48]. When any stress is applied, a decrease in viscosity is observed in systems that show thixotropy behavior, while an increase in viscosity is observed in systems that show rheopectic behavior. After the shear stress is removed in both models, the system slowly returns to its original structures. Rheopexy is more common in colloidal dispersions. The analyses made determined that other formulations except for IS3, IS4, IS5, and IS6 showed rheopectic behavior. However, there is no significant difference between these behavioral models. Since in situ gels are defined as viscous colloidal systems [49], these findings are compatible with the literature [28].

Minitab 18 evaluated the relation between CMAs and CQAs and gave the model summaries. The R2 value demonstrates how the model fits the experimental data, and if the value is close to 100%, the better the model. The adjusted R2 value is the number of terms used in the model considered, and the predicted R2 is an estimation of predicted response values of models are how well [50]. Except for T_sol/gel_ temperature R2, the predicted R2 values were weak or moderate (gelling capacity 0.00%, drug content 26.60%, and log consistency index 51.60%). However, obtained R2 values of CQAs were moderate or strong 97.75% for T_sol/gel_ temperature, 85.88% for log consistency index, 74.81% for drug content, and 53.74% for gelling capacity. These results show that the models obtained are even more successful than the predictions.

Another analysis to understand whether the model is meaningful is the *p* value. Models with a *p* value less than 0.05 are significant [51]. In this study, all models except gelling capacity (*p* value is 0.159) were found to be significant (*p* values< 0.05).

Pareto charts describe the statistical significance of the effects of CMAs on CQAs [52]. When Pareto charts are examined, it is seen that the effect of both Poloxamer types on Tsol/gel and drug content are significant. The impact of Poloxamer 407 on the log consistency index is significant. However, Poloxamer 407 and Poloxamer 188 have no significant effect on gelling capacity.

Characterization studies of the optimized in situ gelling systems (IS-OPT), which we developed using Poloxamer 407 (20% *w*/*v*) and Poloxamer 188 (0.404% *w*/*v*) at the rates determined as a result of the QbD study, were performed. It was observed that IS-OPT was completely clear. The formulation was found to have a drug content of 90.97% and a gelling capacity of “++”. This result is consistent with the gelling capacity of the formulations developed for the QbD study and previous studies. An increase in gelling capacity was observed when poloxamer 407 was used in combination with an appropriate amount of Poloxamer 188 [42]. T_sol/gel_ value was calculated as 31.56 °C. Considering that IS-OPT will dilute to a certain amount with tear film when applied to the eye, the T_sol/gel_ value was found to be 33.91 °C in the analyses made by diluting the formulation in situ gel:STF (50:7). The T_sol/gel_ temperature of IS-OPT, both undiluted and diluted with STF, shows that this formulation will gel at ocular temperatures when applied.

Rheological studies of the optimized formula were carried out not only at 35 °C, but also at 5 and 25 °C. Thus, an idea of the rheological properties of IS-OPT under different storage conditions was obtained. Conversely, since the rheological behaviors may vary with temperature, the optimized formula was evaluated in all aspects. In the rheology examinations, it was determined that IS-OPT showed Herschel-Bulkley flow at all temperatures (5, 25, and 35 °C), thixotropic at 5 °C, and rheopectic at 25 and 35 °C.

Oscillation studies are used to determine the viscoelastic properties of the drugs. It allows for determining the behavior of semisolid dosage forms in the physiological condition after administration by mimicking the physiological environment with the low shear used during analysis. The viscoelastic property affects the ease of application of a drug and its retention time in the application area. If the dose loss is minimized due to an easy administration and the contact time with the target tissue is increased, bioavailability is also observed. For this reason, it is desired that ocular drugs to be applied topically should show certain viscoelastic properties. Examining the viscoelastic property is an important R&D step in selecting the most suitable formulation for clinical use among the new ocular drugs developed [9,28]. Frequency sweep analyses showed G″ ˃ G′ at 5 and 25 °C. These results confirm that IS-OPT was viscoelastic. When the temperature is increased to 35 °C, a rapid increase in the G’ value is observed. This shows that a solid gel structure is formed in the formulation independent of the oscillation frequency. Estimating that IS-OPT will dilute with tear film after it is applied to the eye, oscillation studies were also performed by diluting the formulation in situ gel:STF (50:7) at 35 °C. After dilution with STF, IS-OPT was observed to form a solid gel at 35 °C. However, before dilution, as the oscillation frequency increased, the G′ value, which rose above 10^4^, remained below 10^4^ after dilution.

Texture profile analyses are used in the pharmaceutical field to determine the mechanical properties of dosage forms. It is used primarily for semi-solid drugs to determine parameters such as ease of removal from the package and ability of topical application and to improve the dosage form, depending on these parameters. Conversely, these mechanical properties can be directly related to in vivo sensory parameters [53]. Thus, it is possible to obtain information about the effect of the removal the drug from the package and application comfort in terms of patient compliance, the ability of the drug to spread on the tissue surface, and its effect on elimination and bioavailability.

Hardness, one of the mechanical properties determined during TPA, indicates the force required to deform the gels. This parameter expresses the applicability of the gel to the target tissue [54]. In some studies, it has been stated that hardness can also give an idea about the retention time of the formulation in the application area [53,55]. For a gel to be easily applied, it must have a low hardness value. The hardness value of IS-OPT was found to be low at all three temperatures (0.06 ± 0.00, 0.08 ± 0.00, and 0.16 ± 0.10 N for 5, 25, and 35 °C, respectively). The hardness values obtained at 5 and 25 °C gave preliminary information that the optimized formula can be easily applied to the eye. However, the situation is different at physiological temperatures. Since the formulation will become gel form after application at 35 °C, it is expected that the hardness value will increase. Thus, the drug is expected to remain undiluted and resist nasolacrimal drainage [9]. When the results are examined, it is seen that the hardness value increased at 35 °C. Based on all the results, we can expect that IS-OPT can be easily applied to the eye and will gel after application and will be eliminated more slowly than solution-type preparations.

Compressibility, which defines the work required to compress the product over a given distance, refers to the ease with which the gel is removed from the package and spread at the site of application. The compressibility value should be low such that the gel can be removed from the container and spread easily over the mucosal epithelium [54]. Although the compressibility value of IS-OPT increased with temperature, it was found to be low for all three temperatures (0.26 ± 0.03, 0.39 ± 0.06, and 1.58 ± 0.15 N.mm for 5, 25, and 35 °C, respectively). Compressibility is a parallel value with hardness. For this reason, it is expected to be low at 5 and 25 °C where the hardness is low. However, since the formulation will turn into gel form at 35 °C, its compressibility is expected to increase. Although the compressibility value of IS-OPT increased with temperature, it was also found to be low for 5 and 25 °C. These findings are consistent with previous studies [28].

Adhesiveness is defined as the work required to overcome the attractive forces that occur between the surface of the sample and the surface of the probe during analysis. It represents the adhesion of semi-solid drugs that will occur in vivo conditions to the tissue surface as in vitro conditions. It is thought that with the high adhesiveness value, more drugs will adhere to the tissue surface and will have a longer retention time [54]. In the results, it is seen that the adhesiveness value of IS-OPT increased with temperature (0.10 ± 0.00, 0.20 ± 0.00, and 1.17 ± 0.46 N.mm for 5, 25, and 35 °C, respectively). This shows that the ocular residence time will be long after the in situ gel is applied.

The last parameter measured in TPA, cohesion, represents the restructuring of the gel after application. Cohesion affects the drug’s performance at the application site, and it is expected that more gel will fully recover structurally after application with the higher cohesion value [54]. The cohesion value of IS-OPT increased with temperature (0.57 ± 0.12, 0.73 ± 0.04, and 1.10 ± 0.31 for 5, 25, and 35 °C, respectively). The increase in cohesion value with increasing temperature indicates high restructuring of the optimized in situ gel. According to these results, the performance of IS-OPT in the ocular tissue is expected to be good.

In our previous in vitro release studies, we compared diluted Noxafil^®^ oral suspension with optimized micellar formulation. In our study with Franz diffusion cells, it was seen that there was a dramatic difference between the release profiles of NM-51 and Noxafil due to increasing the aqueous solubility of PSC via micelles [15]. It was seen in rheology and texture analyses that IS-OPT had high hardness at physiological temperatures and formed a solid gel structure. In addition, considering the ocular tear turnover rate, performing in vitro drug release studies on Franz diffusion cells would not be a correct approach, and we worked with a membraneless model [35,36]. We aimed to mimic tear turnover during the study by completely removing the receptor phase and replacing it with a fresh receptor phase at certain time intervals. It was determined that IS-OPT had a cumulative drug release of 71.6352% at the end of the 3rd hour, and the release was in compliance with zero-order kinetics [41]. In situ gels are developed to slow the elimination of topically applied drugs and to increase the drug’s contact time with the ocular tissues. Rheology and texture analyses showed that IS-OPT can contact ocular tissues for a long time, as we aimed. According to the release result, the optimized in situ gel will successfully release the PSC in its content during contact with the ocular tissues and reach the therapeutic dose range.

It was determined that all three samples applied in time-kill assay studies showed anti-fungal activity. At the end of the 24th hour, the log of *C. albicans* surviving in the IS-OPT’s plate was lower than the diluted Noxafil^®^ oral suspension and micelles. When the obtained results were compared with the one-way ANOVA method, it was seen that there was a significant difference between the samples (*p* = 0.0019).

In toxicity studies performed with the HET-CAM method of IS-OPT, no signs of lysis, hemorrhage or coagulation were found in the chorioallantoic membranes at the end of 300 s after applying. This shows us that the ocular application of the optimized in situ gel is safe. In our previous HET-CAM analysis, we compared the micellar formulation and diluted Noxafil^®^ oral suspension. In studies, Noxafil^®^ was found to be slightly toxic with a score of 1.5 [16]. These obtained data question the safety of Noxafil^®^, which is used off-label in the treatment of severe ocular fungal infections in the clinic. For this reason, an ocular dosage form containing PSC as an alternative to Noxafil^®^ is needed in the clinic. Micelles and micellar-based in situ gelling system, which we have optimized in our studies, can be an important alternative.

## 5. Conclusions

Poloxamer 407 is an important excipient for the development of an in situ gel with ideal properties as a thermosensitive copolymer. For this reason, we preferred to use poloxamer 407 while developing the PSC-loaded micellar-based ocular in situ gelling system. However, since the T_sol/gel_ value of poloxamers 407 is low, we used it in combination with poloxamer 188 such that the developed formulations become gel at the natural temperature of the eye. We realized our formulation development studies with the QbD approach and reached the optimized formulation. Characterization studies of IS-OPT have shown that optimized formulation had the rheological and mechanical properties desired for in situ gels for ocular application. In addition, in vitro release and time-kill assay studies show that after IS-OPT is applied to the eye, sufficient PSC release may occur during the ocular retention time, and anti-fungal activity will be observed. HET-CAM toxicity tests have proven that the ocular application of the optimized in situ gel is safe. IS-OPT, which we have optimized in light of these obtained data, may increase the ocular bioavailability of PSC and may be an important alternative to Noxafil^®^ oral suspension, which is used off-label in the clinic. However, it should not be forgotten that these studies are the result of in vitro analyses and should be supported by ex vivo and in vivo analyses.

## Figures and Tables

**Figure 1 pharmaceutics-14-00526-f001:**
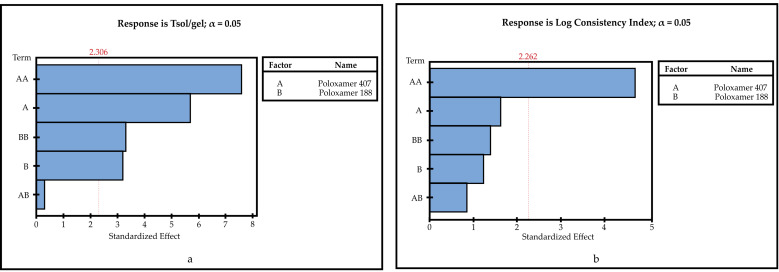
Pareto charts. (**a**) T_sol/gel_; (**b**) log consistency index.

**Figure 2 pharmaceutics-14-00526-f002:**
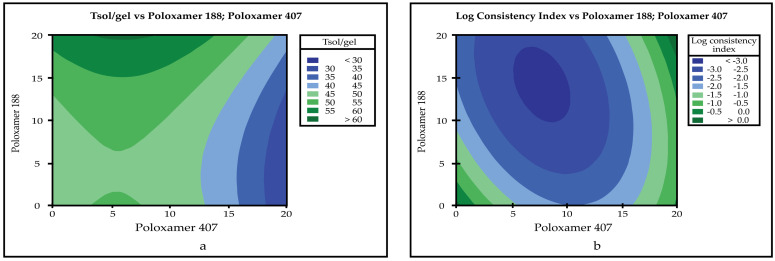
Two-dimensional contour plot graphs; (**a**) T_sol/gel_; (**b**) log consistency index.

**Figure 3 pharmaceutics-14-00526-f003:**
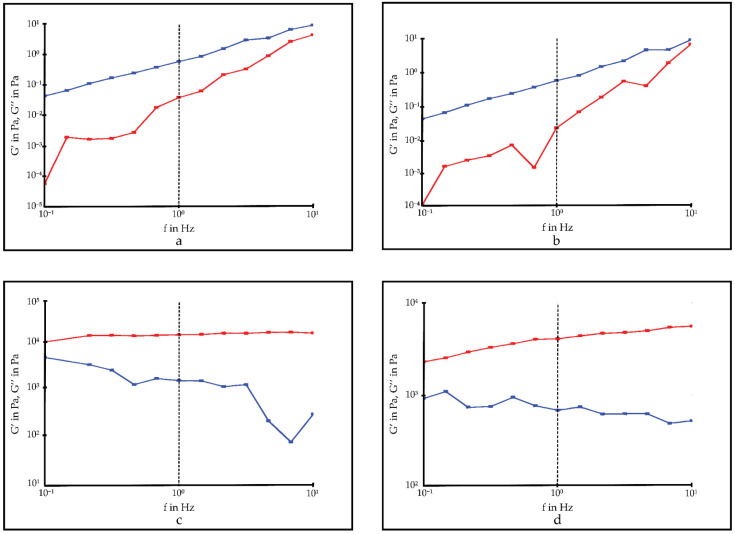
Elastic (G′-red line) and viscous (G″-blue line) modulus as a function of the frequency of IS-OPT. (**a**) 5 °C; (**b**) 25 °C; (**c**) 35 °C; (**d**) in situ gel:STF(50:7) 35 °C.

**Figure 4 pharmaceutics-14-00526-f004:**
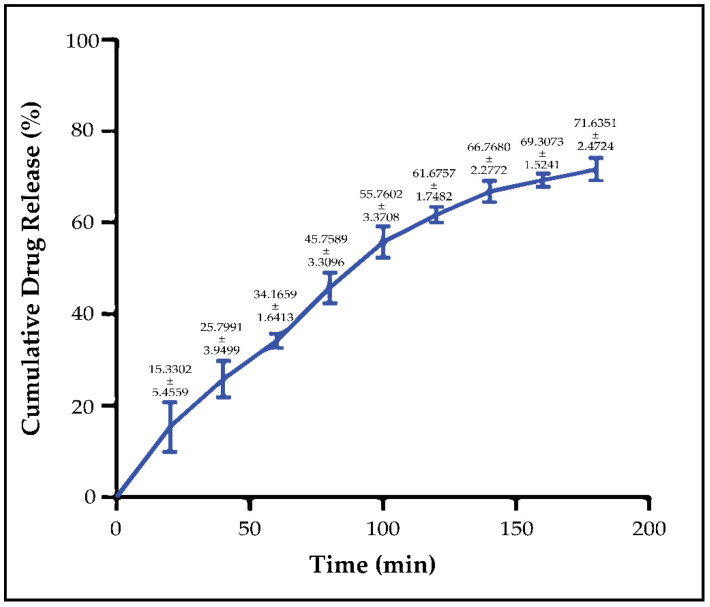
In vitro release value (mean ± stdeva) profile of PSC from the micellar-based in situ gelling system (*n* = 6).

**Figure 5 pharmaceutics-14-00526-f005:**
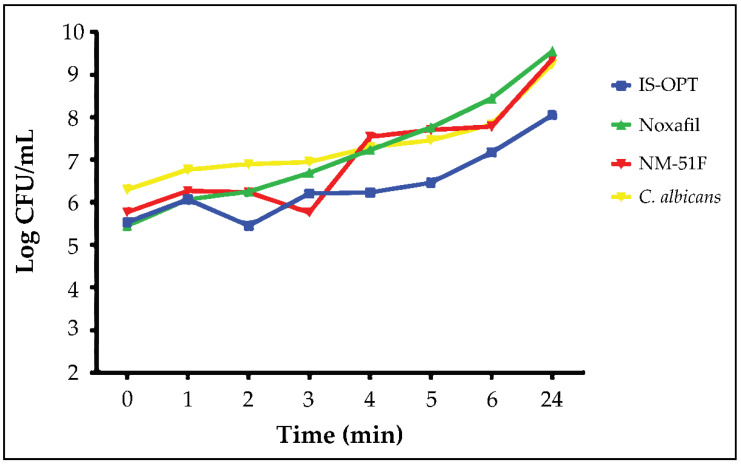
Time-kill determinations against *C. albicans* strain after treatment with diluted Noxafil^®^ oral suspension, micellar-based in situ gelling system, and micelle and *C. albicans* as control. The *x*-axis represents the killing time (h), and the *y*-axis represents the logarithmic *C. albicans* survival (CFU).

**Figure 6 pharmaceutics-14-00526-f006:**
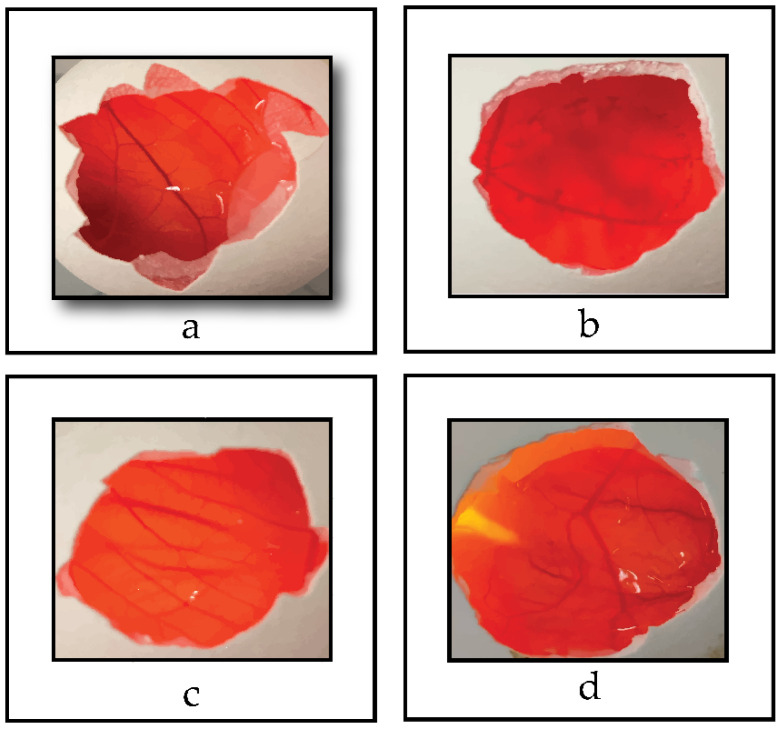
HET-CAM assay; (**a**) negative control, (**b**) positive control, (**c**) IS-OPT Placebo, and (**d**) IS-OPT (*n* = 6).

**Table 1 pharmaceutics-14-00526-t001:** Gelling capacity classification [8].

Gelling Capacity	Definition
-	No gelling
+	Gelation immediately and remained for several minutes
++	Gelation immediately and remained for several hours
+++	Gelation immediately and remained for long hours
++++	Solid gel structure

**Table 2 pharmaceutics-14-00526-t002:** Quality target product profile (QTPP).

Factor	Target	Justification
Route of administration	Ocular	Topical application to the targeted tissue
Delivery system	Micellar-based in situ gels	Increase the contact time of drug delivery system with ocular tissues
Drug content	≥90%	To obtain treatment dose of the API
pH	6.6–7.8	For maximum comfort and patient compliance
Clarity	Clear	Increase the patient compliance
Gelling capacity	Gelation immediately and remained for several or long hours	Affect the contact time of drug delivery system with ocular tissues
T_sol/gel_ temperature	30–35 °C	Transform from solution to gel form at physiological ocular temperature
Log consistency index	0–1	For ease of application and reduction of elimination rate

**Table 3 pharmaceutics-14-00526-t003:** Design of experiment inputs.

Critical Material Attributes	Levels (*w*/*v*%)			
X1: Poloxamer 188	0	15	17.5	20
X2: Poloxamer 47	0	15	17.5	20

**Table 4 pharmaceutics-14-00526-t004:** Scoring scheme for the HET-CAM test for membrane irritation.

	Score		
Effect	30 s	120 s	300 s
Lysis	5	3	1
Hemorrhage	7	5	3
Coagulation	9	7	5

**Table 5 pharmaceutics-14-00526-t005:** Classification scheme for cumulative scores in the HET-CAM test.

Cumulative Score	Irritation Assessment
Up to 0.9	Practically none
1–4.9	Slight
5–8.9	Moderate
9 and above	Strong

**Table 6 pharmaceutics-14-00526-t006:** Polymer ratios and results of pre-formulation studies.

Formulation	Poloxamer 407	Poloxamer 188	HPMC 50M	HPMC 60M	HPMC 75HD100	Carbopol 980	MC	Na CMC	Results
1	15	-	-	-	-	-	-	-	Gelation observed
2	17.5	-	-	-	-	-	-	-	Gelation observed
3	20	-	-	-	-	-	-	-	Gelation observed
4	-	15	-	-	-	-	-	-	Gelation observed
5	-	17.5	-	-	-	-	-	-	Gelation observed
6	-	20	-	-	-	-	-	-	Gelation observed
7	15	15	-	-	-	-	-	-	Gelation observed
8	15	17.5	-	-	-	-	-	-	Gelation observed
9	15	20	-	-	-	-	-	-	Gelation observed
10	17.5	15	-	-	-	-	-	-	Gelation observed
11	17.5	17.5	-	-	-	-	-	-	Gelation observed
12	17.5	20	-	-	-	-	-	-	Gelation observed
13	20	15	-	-	-	-	-	-	Gelation observed
14	20	17.5	-	-	-	-	-	-	Gelation observed
15	20	20	-	-	-	-	-	-	Gelation observed
16	15	-	0.5	-	-	-	-	-	Gelation observed
17	15	-	0.7	-	-	-	-	-	Gelation observed
18	15	-	1	-	-	-	-	-	Gelation observed
19	15	-	-	0.5	-	-	-	-	Gelation observed
20	15	-	-	0.7	-	-	-	-	Gelation observed
21	15	-	-	1	-	-	-	-	Gelation observed
22	15	-	-	-	0.5	-	-	-	No gelation
23	15	-	-	-	0.7	-	-	-	No gelation
24	15	-	-	-	1	-	-	-	No gelation
25	17.5	-	0.5	-	-	-	-	-	Gelation observed
26	17.5	-	0.7	-	-	-	-	-	Gelation observed
27	17.5	-	1	-	-	-	-	-	Gelation observed
28	17.5	-	-	0.5	-	-	-	-	Gelation observed
29	17.5	-	-	0.7	-	-	-	-	Gelation observed
30	17.5	-	-	1	-	-	-	-	Gelation observed
31	17.5	-	-	-	0.5	-	-	-	No gelation
32	17.5	-	-	-	0.7	-	-	-	No gelation
33	17.5	-	-	-	1	-	-	-	No gelation
34	20	-	0.5	-	-	-	-	-	Gelation observed
35	20	-	0.7	-	-	-	-	-	Gelation observed
36	20	-	1	-	-	-	-	-	Gelation observed
37	20	-	-	0.5	-	-	-	-	Gelation observed
38	20	-	-	0.7	-	-	-	-	Gelation observed
39	20	-	-	1	-	-	-	-	Gelation observed
40	20	-	-	-	0.5	-	-	-	No gelation
41	20	-	-	-	0.7	-	-	-	No gelation
42	20	-	-	-	1	-	-	-	No gelation
43	-	15	0.5	-	-	-	-	-	Gelation observed
44	-	15	0.7	-	-	-	-	-	Gelation observed
45	-	15	1	-	-	-	-	-	Gelation observed
46	-	15	-	0.5	-	-	-	-	Gelation observed
47	-	15	-	0.7	-	-	-	-	Gelation observed
48	-	15	-	1	-	-	-	-	Gelation observed
49	-	15	-	-	0.5	-	-	-	No gelation
50	-	15	-	-	0.7	-	-	-	No gelation
51	-	15	-	-	1	-	-	-	No gelation
52	-	17.5				-	-	-	No gelation
53	-	17.5	0.5	-	-	-	-	-	No gelation
54	-	17.5	0.7	-	-	-	-	-	No gelation
55	-	17.5	1	-	-	-	-	-	Gelation observed
56	-	17.5	-	0.5	-	-	-	-	Gelation observed
57	-	17.5	-	0.7	-	-	-	-	Gelation observed
58	-	17.5	-	1	-	-	-	-	Gelation observed
59	-	17.5	-	-	0.5	-	-	-	No gelation
60	-	17.5	-	-	0.7	-	-	-	No gelation
61	-	20	-	-	1	-	-	-	No gelation
62	-	20	0.5	-	-	-	-	-	Gelation observed
63	-	20	0.7	-	-	-	-	-	Gelation observed
64	-	20	1	-	-	-	-	-	No gelation
65	-	20	-	0.5	-	-	-	-	Gelation observed
66	-	20	-	0.7	-	-	-	-	Gelation observed
67	-	20	-	1	-	-	-	-	No gelation
68	-	20	-	-	0.5	-	-	-	No gelation
69	-	20	-	-	0.7	-	-	-	No gelation
70	-	20	-	-	1	-	-	-	No gelation
71	15	-	-	-	-	0.05	-	-	No gelation
72	15	-	-	-	-	0.1	-	-	No gelation
73	15	-	-	-	-	0.15	-	-	No gelation
74	15	-	-	-	-	0.2	-	-	No gelation
75	15	-	-	-	-	0.25	-	-	No gelation
76	-	15	-	-	-	0.05	-	-	No gelation
77	-	15	-	-	-	0.1	-	-	No gelation
78	-	15	-	-	-	0.15	-	-	No gelation
79	-	15	-	-	-	0.2	-	-	No gelation
80	-	15	-	-	-	0.25	-	-	No gelation
81	15	-	-	-	-	-	0.2	-	No gelation, not clear
82	15	-	-	-	-	-	0.5	-	No gelation, not clear
83	15	-	-	-	-	-		0.4	No gelation, not clear
84	-	15	-	-	-	-	0.2	-	No gelation, not clear
85	-	15	-	-	-	-	0.5	-	No gelation, not clear
86	-	15	-	-	-	-	-	0.4	No gelation, not clear

**Table 7 pharmaceutics-14-00526-t007:** Polymer ratios, pH values, clarity, gelling capacity, and drug content of in situ gels.

Formulation	Poloxamer 407	Poloxamer 188	Clarity	pH	Gelling Capacity	Drug Content
IS1	15	-	Clear	7.4	-	100.40
IS2	17.5	-	Clear	7.4	+	99.47
IS3	20	-	Clear	7.35	+	97.48
IS4	-	15	Clear	7.4	-	100.13
IS5	-	17.5	Clear	7.41	-	94.56
IS6	-	20	Clear	7.45	-	92.1
IS7	15	15	Clear	7.34	-	96.34
IS8	15	17.5	Clear	7.37	-	98.53
IS9	15	20	Clear	7.35	-	91.22
IS10	17.5	15	Clear	7.4	-	94.56
IS11	17.5	17.5	Clear	7.41	-	92.1
IS12	17.5	20	Clear	7.39	-	94.58
IS13	20	15	Clear	7.4	++	93.07
IS14	20	17.5	Clear	7.4	-	91.73
IS15	20	20	Clear	7.41	-	93.21
IS-OPT	20	0.404	Clear	7.39	++	90.97

**Table 8 pharmaceutics-14-00526-t008:** The rheological properties of in situ gels in the range between IS1-IS15 containing different ratios of Poloxamer 407/188 at 35 °C (*n* = 3).

Formulation	Poloxamer 407	Poloxamer 188	Ʈ_0_ (Pa)	K (Pa.s)	η	Hysteresis Area	T_sol/gel_
**IS1**	15	-	1.5613 ± 0.1617	0.0411 ± 0.0181	1.0378 ± 0.0621	−2955.00 ± 1132.06	42.72 ± 1.23
**IS2**	17.5	-	153.5667 ± 18.0514	0.0209 ± 0.01275	1.1340 ± 0.0827	−28,052.33 ± 23,665.50	35.75 ± 0.03
**IS3**	20	-	204.0667 ± 15.0699	0.7253 ± 0.5412	0.9527 ± 0.3337	3117.67 ± 9595.36	30.36 ± 0.27
**IS4**	-	15	0.3178 ± 0.3340	0.0065 ± 0.0032	1.0547 ± 0.0992	2216.33 ± 773.13	50.62 ± 0.92
**IS5**	-	17.5	0.4035 ± 0.1407	0.0068 ± 0.0019	1.0537 ± 0.0487	1446.67 ± 542.53	55.22 ± 3.9
**IS6**	-	20	0.4759 ± 0.0628	0.0098 ± 0.0011	1.0480 ± 0.0154	675.73 ± 278.41	-
**IS7**	15	15	3.9093 ± 0.6473	0.0223 ± 0.0063	1.1787 ± 0.0290	−27,526.67 ± 13,332.85	46.85 ± 0.47
**IS8**	15	17.5	6.9697 ± 1.2405	0.0142 ± 0.0019	1.2807 ± 0.0265	−34,804.33 ± 38,112.95	50.57 ± 1.73
**IS9**	15	20	18.2967 ± 2.7344	0.0119 ± 0.0058	1.4470 ± 0.0183	−12,055.00 ± 2156.67	51.95 ± 0.04
**IS10**	17.5	15	2.9977 ± 2.5657	0.0854 ± 0.0252	1.0650 ± 0.0702	−28,046.67 ± 14,896.65	41.00 ± 0.06
**IS11**	17.5	17.5	5.4685 ± 4.9357	0.1174 ± 0.0968	1.1000 ± 0.1293	−139,000.00 ± 37,070.74	46.47 ± 0.4
**IS12**	17.5	20	12.7900 ± 2.5590	0.0796 ± 0.0225	1.1993 ± 0.0176	−234,866.67 ± 118,235.75	48.18 ± 3.41
**IS13**	20	15	−1.6990 ± 7.1550	0.2297 ± 0.1736	1.0047 ± 0,1353	−179,730.00 ± 134,857.11	35.51 ± 0.21
**IS14**	20	17.5	−2.1033 ± 15.5105	0.6272 ± 0.4439	0.9414 ± 0.1520	−179,033.33 ± 61,502.87	40.94 ± 0.09
**IS15**	20	20	−107.4100 ± 108.47959	13.6847 ± 12.5832	0.6972 ± 0.3390	−249,233.33 ± 68,141.05	41.28 ± 0.28

**Table 9 pharmaceutics-14-00526-t009:** Obtained R^2^ values.

CQA	R^2^	Adjusted R^2^	Predicted R^2^
T_sol/gel_ temperature	97.75%	96.34%	91.66%
Gelling capacity	53.74%	28.05%	0.00%
Drug content	74.81%	60.81%	26.60%
Log consistency index	85.88%	78.04%	51.60%

**Table 10 pharmaceutics-14-00526-t010:** Obtained *p* values.

CQA	*p* Value
T_sol/gel_ temperature	0.000
Gelling capacity	0.159
Drug content	0.015
Log consistency index	0.001

**Table 11 pharmaceutics-14-00526-t011:** The rheological properties of optimized in situ gel (IS-OPT) at 5, 25 and 35 °C (*n* = 3).

Temperature	Shear Rate (Ʈ_0_-Pa)	Consistency Index (k-Pa.s)	Rheological Exponent	Hysteresis Area
5 °C	0.2945 ± 0.2752	0.06332 ± 0.05627	0.9411 ± 0.0830	8025 ± 5045.87
25 °C	1.3503 ± 0.6372	0.2692 ± 0.0459	0.9490 ± 0.0239	−18,336.67 ± 6730.53
35 °C	172.47 ± 75.2095	0.5174 ± 0.5842	0.8521 ± 0.1809	−12,080.67 ± 5971.31

**Table 12 pharmaceutics-14-00526-t012:** The mechanical properties of IS-OPT at 5, 25, 35 °C (*n* = 3).

Temperature	Hardness (N)	Compressibility (N.mm)	Adhesiveness (N.mm)	Cohesiveness
5 °C	0.06 ± 0.00	0.26 ± 0.03	0.10 ± 0.00	0.57 ± 0.12
25 °C	0.08 ± 0.00	0.39 ± 0.06	0.20 ± 0.00	0.73 ± 0.04
35 °C	0.16 ± 0.10	1.58 ± 0.15	1.17 ± 0.46	1.10 ± 0.31

## Data Availability

Data supporting reported results will be shared openly when the corresponding author is contacted.

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
