# Peer review of "Optimization of the Micellar-Based In Situ Gelling Systems Posaconazole with Quality by Design (QbD) Approach and Characterization by In Vitro Studies"

_pharmaceutics, 2022, doi:10.3390/pharmaceutics14030526_

Round 1

Reviewer 1 Report

General comments: Durgun et al., are proposing a micellar based in situ gelling hydrogel for the delivery of posaconazole in the context of ocular fungal infections. The work claims to use a Quality-by-design approach and a deep in vitro characterization from the rheological standpoint. The findings are interesting with a candidate hydrogel that gels in situ at physiological temperature but is in solution state at “shelf” temperature, that bring pharmaceutical value. The release of posaconazole is studied overtime, and a study of toxicity is performed on chorioallantoic membranes. Overall, a thorough English editing needs to be performed as it makes the reading difficult and impairs both the delivered message and the readers understanding. Some words are missing and typos are found throughout the text.

Comment 1: It is mentioned throughout the paper that a QbD approach has been performed, however the QTTP have not been described precisely, the CQA have been defined but not explained for example in what regards does the Gelling capacity have an impact on the clinical application, and what would be your targets. This would really improve the QbD dimension of the paper, as it is mentioned in the title of the paper.

Comment 2: The terms QbD and Design of experiments are a bit mixed up in the paper. It is also mentioned at the line 191 that a mixture design methodology was employed, but the table 2 and table 6 seems to refer to an OFAT (one factor at the time). Could you explain? Also, the equilibrium of the design is somehow skewed toward the values of 15; 17.5; and 20% w/v so the optimal concentration of Poloxamer 47 being at 0.404 might be off regarding the unknown response surface between 0 and 15% w/v. In that regards, more details on the methodology of Minitab should be discussed here.

Comment 4: for example the model fitted through the points is not explicit (even though some principal, interaction and squared parameters seem to be involved in fig 1). So concluding on the r² without having the equations does not make much sense, and overfitting might be possible. On what grounds is the optimal formulation is selected? Without all of that, interpretation and reproducibility of the experiments is difficult.

Comment 5: On figure 3 it is not clear which color corresponds to which modulus, is it possible to add a legend to facilitate the lecture?

Comment 6: On figure 4, why is the release not reaching 100% at 3 hours if all the gel is dissolved? Could you provide an explanation

Comment 7: Have you verified the effective release from the gel, are those micelles released, the free drug or a combination of both?

Extra details on the typos:

Title – There is a typo “o posaconazole”, “(QbD)Approach”

Line 40 – that have been shown

Line 41 – “That is”, “in the commercial product” needs English enditing

Line 44 – “hydropholic”

Line 48 – “its preparation”

Line 54 – impair instead of prevent

Line 56 – “PSC is the triazole group” something is missing in the sentence

Line 64 – extra “r” after the reference

Line 71-73 – Please reformulate the sentence to make it clearer

Line 89 – Please define QTPP, CQA  

Line 95 – Throughout the text please pay attention to italicize the latin words

Material and methods : Please keep remain consistent while describing the products (Supplier, city, state, country)

Line 118 : Remove the “-“ after 0.45

Line 254 : “IS-OPT” not described before

Line 260 : “TexturePro CT V1.6 Build software” mention supplier please

Line 288 : “thegels” there is no space between the two words

Line 294 : “C. albi-cans” remove the dash in the word

Line 300 : “3,4” add an extraspace after the comma before the 4

Line 304 : Put the 1 in words : one log 10 CFU/ml it will add readability

Line 333 : “2.11. StatisticalAnalysis”, please add a space between the two words

Line 348 : “50 M HPMC” please remove the space between 50 and M

Line 350 – 353 : Please reformulate the sentence is hard to understand, plus this mentions a scale-up that is nowhere to be found

Table 6 : There is a typo “geling capacity”, please add a unit to drug content to add understanding

Figure 4 : Please add the value observed (e.g. mean +/- stdeva)

Line 516 : A scale up is mentioned here that does not happen explicitly in the text

Line 522 : The sentence is hard to understand could you please reformulate

Line 618 : However is written twice

Author Response

RESPONSES TO THE COMMENTS OF REVIEWER 1

We thank Reviewer for her/his valuable comments and suggestions that helped improve the quality of our manuscript. The following responses were prepared to address all of the reviewers’ comments in a point-by-point manner.

General Comments: Durgun et al., are proposing a micellar based in situ gelling hydrogel for the delivery of posaconazole in the context of ocular fungal infections. The work claims to use a Quality-by-design approach and a deep in vitro characterization from the rheological standpoint. The findings are interesting with a candidate hydrogel that gels in situ at physiological temperature but is in solution state at “shelf” temperature, that bring pharmaceutical value. The release of posaconazole is studied overtime, and a study of toxicity is performed on chorioallantoic membranes. Overall, a thorough English editing needs to be performed as it makes the reading difficult and impairs both the delivered message and the readers understanding. Some words are missing and typos are found throughout the text.    

Response: Thank you for your comment and encouragement that our manuscript is promising. We are sorry about our typographical errors and grammatical mistakes. Since we are not native English, we predicted that we could make grammatical mistakes and have the whole article checked by the Grammarly program. This program is used by The New York Times, Wall Street Journal, Forbes, USA Today, and TechCrunch. It is also recommended for use in the academic field in the USA. However, we will still review the grammar of the manuscript. The language revision and edition of the text were conducted according to your warnings. If you have any problems related to language revision, please let us know.

Specific Comments:

Comment 1.

“It is mentioned throughout the paper that a QbD approach has been performed, however the QTTP have not been described precisely, the CQA have been defined but not explained for example in what regards does the Gelling capacity have an impact on the clinical application, and what would be your targets. This would really improve the QbD dimension of the paper, as it is mentioned in the title of the paper.”

Response 1.

Thank you for your comments. In the relevant sections throughout the manuscript, we have stated why the QTPPs in the QbD study are important for ocular in situ gels. However, according to your request, we have added the explanations and "Table 2. Quality Target Product Profile (QTPP)", in which we summarize definitions of QTPPs, to the "2.5.In situ Gel Formulation Development Based on the Quality by Design and Optimization" section. For gelling capacity, we explained why it is a critical parameter for gels and evaluated our results in the eighth paragraph of the discussion section on lines 540-567. Therefore, we did not make any further additions to this section. However, if this explanation is not sufficient for you, please inform us.

Comment 2.

“The terms QbD and Design of experiments are a bit mixed up in the paper. It is also mentioned at the line 191 that a mixture design methodology was employed, but the table 2 and table 6 seems to refer to an OFAT (one factor at the time). Could you explain? Also, the equilibrium of the design is somehow skewed toward the values of 15; 17.5; and 20% w/v so the optimal concentration of Poloxamer 47 being at 0.404 might be off regarding the unknown response surface between 0 and 15% w/v. In that regards, more details on the methodology of Minitab should be discussed here.”

Response 2.

Thank you for your comments. Table 3 (old table 2) belongs to the pre-formulation studies, and this part is shared for general information purposes only. The experimental design has not been applied. The sections related to the QbD study are "2.5.In situ Gel Formulation Development Based on the Quality by Design and Optimization" and "3.3.In situ Gel Formulation Development Based on the Quality by Design and Optimization". In section 2.5,  A three-level full factorial design was applied, on the other hand, to see just one surfactant's effects on the formulations, the formulations were produced containing only 1 surfactant. And this design was named a mixture design because it did not fall into a specific group. An explanation of the concentration recommended by the program for Poloxamer 188 has been added in the section 3.3   “Although there is no data for 0.404 % (w/v) in formulation studies for Poloxamer 188, the program has determined this concentration by evaluating formulations without Poloxamer 188.”

Comment 3.

“for example the model fitted through the points is not explicit (even though some principal, interaction and squared parameters seem to be involved in fig 1). So concluding on the r² without having the equations does not make much sense, and overfitting might be possible. On what grounds is the optimal formulation is selected? Without all of that, interpretation and reproducibility of the experiments is difficult.”

Response 3.

Thank you for your comments. Since the article was not prepared by focusing on algorithm development, the equation data were not included in the article. However, the equations of the studied data are listed below;

Gelling capacity= -0,82+0,033 Poloxamer 407+0,208 Poloxamer188 + 0,00618 Poloxamer407 * Poloxamer 407-0,00586 Poloxamer 188* Poloxamer188-0,00742 Poloxamer 407*Poloxamer188

Tsol/gel= 47,77+ 1,004 Poloxamer 407- 0,379 Poloxamer 188 – 0,0938 Poloxamer 407* Poloxamer 407 + 0,0426 Poloxamer 188* Poloxamer 188+ 0,0063 Poloxamer 407* Poloxamer 188

Drug Content= 113,84-0,570Poloxamer 407-0,876 Poloxamer188 – 0,0153 Poloxamer 407*Poloxamer 407-0,0094 Poloxamer 188* Poloxamer 188+ 0,0426 Poloxamer 407*Poloxamer 188

Log consistency index= 0,12-0,395 Poloxamer 407-0,240 Poloxamer 188+ 0,01845 Poloxamer 407* Poloxamer 407 + 0,00633 Poloxamer 188* Poloxamer 188+ 0,00780 Poloxamer 407* Poloxamer 188

Comment 4.

On figure 3 it is not clear which color corresponds to which modulus, is it possible to add a legend to facilitate the lecture?”

Response 4.

We apologize for this problem. We increased the resolution of all figures in the manuscript to avoid any clarity issues. After this change, when we view the figures separately and in the manuscript, we do not experience any resolution problems. However, if you have any problems, please let us know.

Comment 5.

“On figure 4, why is the release not reaching 100% at 3 hours if all the gel is dissolved? Could you provide an explanation.”

Response 5.

Thank you for your comment. The formulations developed in this study are not single in situ gel formulations. Micelles, which we optimized before, were gelled in situ using Poloxamer 407 and/or Poloxamer 188. For this reason, the release of Posaconazole occurs in two stages. Our previous studies determined that the release of posaconazole from the micelles continued for 7 hours and reached the maximum value (118.95±1.60 μg/cm2 = approximately 64%) at the end of this period (DOI: 10.2174/1381612826666200313172207). In this study, we found that the release of Posaconazole was 71.6352% when the optimized in situ gel (IS-OPT) was completely dissolved in simulated tear fluid (STF) in 3 hours. In fact, cumulative Posaconazole release is consistent with our previous study. We think that the observed time and 7% difference in reaching the maximum value is due to the sink conditions. The previous study was performed in Franz Diffusion cells using 12 mL of receptor phase. In the release study of micellar-based in situ gels, the 2 mL receptor phase was used and was refreshed every 20 minutes, and a total of 20 mL receptor phase was used for each group. This has undoubtedly changed the sink condition. On the other hand, this change made every 20 minutes is consistent with the tear turnover rate and amount (tear turnover rate 0.5-2.2 µl/minute). For this reason, we think that this in vitro method, in which we performed the release studies of micellar-based in situ gels, mimics physiological conditions and provides reliable information about the in vivo fate of in formulation.

Although the in vitro release study was performed with 2 mL, in practice a drop is approximately 50 µL. Because the formulation contains 250 µg/mL API, 12.5 µg Posaconazole will be administered to the patient in once. Considering that the in situ gel will remain insoluble for 3 hours and a 70% release will occur, 8.75 µg of Posaconazole will come into contact with ocular tissues. European Committee on Antimicrobial Susceptibility Testing (EUCAST) declared minimum inhibitory concentration (MIC) of PSC for susceptible C. albicans < 0.06 μg/mL and resistant C. albicans > 0.06 μg/mL (DOI: 10.1111/j.1469-0691.2011.03646.x.). Considering the obtained data and mentioned technical note, it could be suggested that PSC-loaded micellar-based in situ gel (250 μg/mL) could significantly inhibit C. albicans in ocular anti-fungal diseases.

Comment 6.

“Have you verified the effective release from the gel, are those micelles released, the free drug or a combination of both?”

Response 6.

Thank you for your comments. As we stated in Response 5, the results obtained are consistent with the cumulative release of Posaconazole from the micelles. In the in situ gel formulations produced in this study, all of the Posaconazole is contained in the micelles. The presence of free Posaconazole is not possible. There are two ways we can be sure that there is no free drug:

  1. The particle size of the Posaconazole used is approximately 1400 nm (Sharpe S, Sequeira J, Harris D, Shashank M. Antifungal Composition with enhanced Bioavailability. US 8.263,600 B2, 2012.). The final filtration process while producing micelles was carried out on 0.45 µm PTFE filters. If there is free Posaconazole remaining without being loaded into the micelle, it will be retained in the filter during this process.
  2. Posaconazole is practically insoluble in water. Methanol is used in the preparation of micelles to dissolve it. After the organic solvent evaporation, the micelles are hydrated. Our previous studies found that Posaconazole, which was not loaded into the micelles, precipitated during this hydration process (DOI: 10.2174/1381612826666200313172207). However, such a situation was not observed in the micelle formulation used in the production of in situ  

Comment 7.

“Extra details on the typos:”

Title – There is a typo “o posaconazole”, “(QbD)Approach”

Line 40 – that have been shown

Line 41 – “That is”, “in the commercial product” needs English enditing

Line 44 – “hydropholic”

Line 48 – “its preparation”

Line 54 – impair instead of prevent

Line 56 – “PSC is the triazole group” something is missing in the sentence

Line 64 – extra “r” after the reference

Line 71-73 – Please reformulate the sentence to make it clearer

Line 89 – Please define QTPP, CQA 

Line 95 – Throughout the text please pay attention to italicize the latin words

Material and methods : Please keep remain consistent while describing the products (Supplier, city, state, country)

Line 118 : Remove the “-“ after 0.45

Line 254 : “IS-OPT” not described before

Line 260 : “TexturePro CT V1.6 Build software” mention supplier please

Line 288 : “thegels” there is no space between the two words

Line 294 : “C. albi-cans” remove the dash in the word

Line 300 : “3,4” add an extraspace after the comma before the 4

Line 304 : Put the 1 in words : one log 10 CFU/ml it will add readability

Line 333 : “2.11. StatisticalAnalysis”, please add a space between the two words

Line 348 : “50 M HPMC” please remove the space between 50 and M

Line 350 – 353 : Please reformulate the sentence is hard to understand, plus this mentions a scale-up that is nowhere to be found

Table 6 : There is a typo “geling capacity”, please add a unit to drug content to add understanding

Figure 4 : Please add the value observed (e.g. mean +/- stdeva)

Line 516 : A scale up is mentioned here that does not happen explicitly in the text

Line 522 : The sentence is hard to understand could you please reformulate

Line 618 : However is written twice

Response 7.

Thank you for your comment, and we are sorry about our typographical errors and grammatical mistakes. As we stated in our response to your general comment, since we are not native English, we predicted that we could make grammatical mistakes and have the whole article checked by the Grammarly program. The language revision and edition of the text were conducted according to your warnings. If you have any problems related to language revision, please let us know. Special response for comment "Line 516 : A scale up is mentioned here that does not happen explicitly in the text": Pre-formulation studies were carried out on 5 mL samples. It defines productions of 100 mL, which we refer to as scale-up. However, we do not think that it is necessary to give such detailed information.

Reviewer 2 Report

The manuscript entitled “Optimization of the Micellar Based In situ Gelling Systems of Posaconazole with Quality by Design (QbD)Approach and Characterization by In Vitro Studies” deals with a mainly rheological characterization of a previously developed TPGS micelles embedded in an in situ thermogelling system. The manuscript contains large amount of experimental results, however significance of content is not completely clear. In present form the manuscript is not suitable for publication, a significant revision in methodology and presentation of results are required to clarify the following issues:

The authors mention the manuscript deals with a QbD driven optimization process of micellar-based in situ gelling ocular system, however it is not clear how the determination of CQAs, CMAs and QTPPs was acquired. At least in the supplementary date an Ishikawa diagram or the severity scores of these QbD parameters should be presented to make clear the selection of factors was objective independently from authors own opinion.

I found a lot of similarities with a previously published paper of the authors (Durgun, M.E.; Kahraman, E.; Güngör, S.; Özsoy, Y. Optimization and Characterization of Aqueous Micellar Formulations 818 for Ocular Delivery of an Antifungal Drug, Posaconazole. Curr. Pharm. Des. 2020, 26, 1543–1555, 819 doi:10.2174/1381612826666200313172207), which makes me the novelty of present study questionable.

The authors claim methanol and acetonitrile was applied during the preparation of micelles. I wonder whether the authors did check residual solvent content in the optimized formulation?

The authors applied TPGS as a good polymeric micelle former material for solubilizing the drug, however no investigation on micellar characteristics was carried out. This should be further investigated.

In line 127, the authors claim “the effect of different polymer types on the formulation was investigated”. What kind of investigations were carried out? Please add missing information to the method.

Section 2.4.2 and 2.6.2 describes the same procedure. Please combine both methodology, to reduce the spread of manuscript, which is anyway too long.

What kind of Design of Experiment was carried out in the pre-formulation studies? How did the authors get 86 runs of experiment?

Please improve the quality if Fig 1, 3 and 5, the fonts of axes are too small and blurred.

The authors claim the optimized formulation contained 20% of Poloxamer 407 and 0.4% of poloxamer 188. Poloxamer 188 seems to have no effect in reaching the optimal sol-gel transition. What is the reason for applying it in the formulation?

The manuscript presents unnecessary data about other kind of polymers (Carbopol, HPMC) which showed no advantageous properties on the formulation. These unnecessary results could be omitted from manuscript, only in one sentence can be explained that they are not suitable for the purpose.

The discussion section is too long, pleas shorten it by focusing on the most important results of work.

Author Response

RESPONSES TO THE COMMENTS OF REVIEWER 2

We thank Reviewer for her/his valuable comments and suggestions that helped improve the quality of our manuscript. The following responses were prepared to address all of the reviewers’ comments in a point-by-point manner.

General Comments: The manuscript entitled “Optimization of the Micellar Based In situ Gelling Systems of Posaconazole with Quality by Design (QbD)Approach and Characterization by In Vitro Studies” deals with a mainly rheological characterization of a previously developed TPGS micelles embedded in an in situ thermogelling system. The manuscript contains large amount of experimental results, however significance of content is not completely clear. In present form the manuscript is not suitable for publication, a significant revision in methodology and presentation of results are required to clarify the following issues.

Response: Thank you for your comment. Posaconazole is a triazole anti-fungal agent used off-label to treat ocular fungal infections, although it does not have an ocular form due to its wide spectrum. For this purpose, it is used in the clinic as the oral suspension of Noxafil, which is applied topically to the eye after dilution. Based on this knowledge in our previous studies, we developed and optimized Posaconazole-loaded ocular micellar formulations. We have given the reasons for choosing micelles as the carrier system in lines 63-70 in the introduction of this manuscript code pharmaceutics-1574669:

  • “Because of the high lipophilic character of PSC, micelles is a suitable carrier system for the eye that contains hydrophilic and lipophilic tissues together [1],
  • The particle size of the micelles can be adjusted in accordance with the pore size of the ocular tissues [1],
  • The presence of different ocular commercial products with micellar structures in the market [17–20],
  • Micelles can be produced with reliable copolymers such as TPGS that have received GRAS approval [21].”

In recent years, the combined use of drug delivery systems has been a very common practice. Thus, the advantages of different carrier systems can be utilized, and the bioavailability of drugs can be further increased. It is common practice to convert any nanocarrier developed in the ocular field into an in situ gelling systems. Thanks to this application, parameters such as particle size, solubility profile, amount of dose are changed with nanocarriers. On the other hand, the contact time of the formulation with ocular tissues is increased and the elimination rate is reduced with its in situ gel structure. Micellar-based in situ gels are the most commonly used combination model for the combined use of ocular-targeted delivery systems. We mentioned the importance of this combined use in the introduction part of our article, lines 71-85.

This study was carried out in order to increase the contact time of micelles with ocular tissues, which we optimized in our previous studies and proved their efficiency in vitro and ex vivo (doi.org/10.1007/s13346-021-00974-x and doi:10.2174/1381612826666200313172207). We think that the information we have given in the above line spacing is sufficient to state the purpose and importance of our study. We are afraid that giving more detailed information will make the "introduction" part too long. In addition, as we stated in the discussion and conclusion section of the experimental findings, we think that our optimized in situ gel formulation can be an alternative to Noxafil oral suspension, which is used off-label in the clinic.

Specific Comments:

Comment 1.

“The authors mention the manuscript deals with a QbD driven optimization process of micellar-based in situ gelling ocular system, however it is not clear how the determination of CQAs, CMAs and QTPPs was acquired. At least in the supplementary date an Ishikawa diagram or the severity scores of these QbD parameters should be presented to make clear the selection of factors was objective independently from authors own opinion.”

Response 1.

Thank you for your comments. In the relevant sections throughout the manuscript, we have stated why the QTPPs in the QbD study are important for ocular in situ gels. However, according to your request, we have added the explanations and "Table 2. Quality Target Product Profile (QTPP)", in which we summarize definitions of QTPPs, to the "2.5.In situ Gel Formulation Development Based on the Quality by Design and Optimization" section. For gelling capacity, we explained why it is a critical parameter for gels and evaluated our results in the eighth paragraph of the discussion section on lines 540-567. Therefore, we did not make any further additions to this section. However, if this explanation is not sufficient for you, please inform us.

Comment 2.

“I found a lot of similarities with a previously published paper of the authors (Durgun, M.E.; Kahraman, E.; Güngör, S.; Özsoy, Y. Optimization and Characterization of Aqueous Micellar Formulations for Ocular Delivery of an Antifungal Drug, Posaconazole. Curr. Pharm. Des. 2020, 26, 1543–1555, 819 doi:10.2174/1381612826666200313172207), which makes me the novelty of present study questionable.”

Response 2.

Thank you for your comment. We think there is confusion. The previous study, in which you mentioned that there are similarities, is only concerned with the optimization and characterization of Posaconazole-loaded ocular micelles. The in vitro properties of the 4 micelle formulations we selected as a result of preformulation studies were examined in the article In this manuscript with the ID number of “pharmaceutics-1574669”, a single TPGS micelle formulation that we selected according to the findings of the previous study was turned into an in situ gel. The only thing similar to our previous article published with the doi:10.2174/1381612826666200313172207 is the preparation method and characterization of the micelle. These studies are explained in the section “2.1.Preparation and Characterization of PSC-loaded TPGS Micelles” with reference to the previous article.

We think this misunderstanding is caused by the copolymers used in the formulations. In the previous study, micellar formulations were designed as single and mixed micelles. While all four optimized micelle formulations contained TPGS, one of the mixed micelles contained 20 mg/mL Poloxamer 407 in addition to TPGS. The other contained 20 mg/mL Poloxamer 188 in addition to TPGS. However, since the results of these mixed micelles were not considered appropriate, one of the single TPGS micellar formulations was used while preparing the in situ gel. Poloxamers can be used in both micelles and in situ gel formulation. We used Poloxamers in this study because we aimed to develop thermosensitive in situ gels. However, there are differences between the uses of Poloxamer in both studies. We outline these differences below:

  • Poloxamers were used in the amount of 20 mg/mL for micelle preparation. Usage rates for in situ gels are 15-20% (w/v) (150 - 200 mg/mL). These amounts 7.5-10 fold of the amount we use when producing mixed micelles.
  • While producing the micelle, Poloxamers were used to form the micelle structure. It was dissolved in acetonitrile together with TPGS and then mixed with Posaconazole solution to form a micellar structure. Then, the organic solvents were evaporated, and the micelles were turned into a thin film layer. Finally, this thin film layer was hydrated and the micelles were dispersed in water. When preparing in situ gel, Poloxamers were added directly to the micelle dispersion in water. It is not included in the micelle structure.
  • When preparing micelles, TPGS was combined with Poloxamer 407 or 188 separately. In the in situ gels, there are formulations in which both Poloxamer types are used simultaneously.

Comment 3.

“The authors claim methanol and acetonitrile was applied during the preparation of micelles. I wonder whether the authors did check residual solvent content in the optimized formulation?”

Response 3.

Thank you for your comment. When the previous studies on the development of ocular drug delivery systems are examined, it is seen that the dissolution of API and/or copolymers is performed using organic solvents in almost all of them. Residual solvent content was not examined in almost all of these studies. Studies using in vivo animal experiments are also included in this group. In the previous studies where we developed and optimized posaconazole-loaded micelles, we did not perform residual solvent content control, as we only performed in vitro and ex vivo evaluations. However, the results of HET-CAM toxicity tests showed that micelles are suitable for the ocular application. If there was a solvent residue, this situation could be easily observed in HET-CAM studies. In a retrospective study, the results of HET-CAM and Draize tests were compared according to The UN Globally Harmonized System of Classification and Labeling of Chemicals (UN GHS) which are also applied in the European Union (EU GHS) and the European Union’s "Dangerous Substances Directive" criteria (DSD) (doi.org/10.1016/j.yrtph.2011.02.003.). According to the results, the HET-CAM and Draize test results were found to be 80–90% compatible in all pathological cases. For this reason, we can say that the micelles are reliable in terms of solvent residue. However, solvent residue analysis will be checked separately before in vivo studies are performed.

Comment 4.

“The authors applied TPGS as a good polymeric micelle former material for solubilizing the drug, however no investigation on micellar characteristics was carried out. This should be further investigated.”

Response 4.

Thank you for your comment. The subject of this study is not the effect of polymer/surfactants used in the preparation of micelles on the solubility of API or the characterization of micelles. We demonstrated the effect of TPGS in the preparation of Posaconazole-loaded micelles, the potential of optimized micelles to improve the solubility of Posaconazole, and to increase the permeation and penetration of Posaconazole into ocular tissues with the micellar structure in our previous studies by comparing with Noxafil oral suspension, which was used off-label in clinic (doi.org/10.1007/s13346-021-00974-x and doi:10.2174/1381612826666200313172207). This article with the ID number of “pharmaceutics-1574669” focuses on the development of the in situ gelling systems of the micellar formulation, which gave the best results in our previous studies. For this reason, we did not find it necessary to give the parameters of the characterization of the micelle again. If we refer to the effect of TPGS on the preparation of Posaconazole-loaded micelle and the studies on the characterization of the micelles in this manuscript, as you mentioned in comment 2, it is highly similar to our previous studies. This is something we want to avoid.  

Comment 5.

“In line 127, the authors claim “the effect of different polymer types on the formulation was investigated”. What kind of investigations were carried out? Please add missing information to the method.”

Response 5.

Thank you for your comment. The statement “the effect of different polymer types on the formulation was investigated” in line 127 is included in the section "2.2.Pre-formulation Studies of Micellar Based In situ Gelling Systems". In this study, it was aimed to develop a thermosensitive in situ gel. For this purpose, Poloxamer 407 and 188 were used as the main polymer. Other polymers were used in pre-formulation studies to increase the mucoadhesive property. In pre-formulation studies, in situ gels were evaluated only for gelation at any temperature and for clarity. The amount of polymer used was determined in accordance with the literature. Pre-formulation studies were carried out to obtain general information. Formulations that did not show any gelation at any temperature or were not clear were directly screened, and no further studies were conducted with these formulations. Although successful results were obtained at first in formulations where poloxamers were combined with HPMC 50M and HPMC 60M, aggregation problems arose when they were produced on a large scale. For this reason, these formulations were also eliminated. The purpose of using other polymers, how the pre-formulation study is performed, and how the formulations to be used in the QbD study are determined in detail in lines 463-520 in the discussion section.

Comment 6.

“Section 2.4.2 and 2.6.2 describes the same procedure. Please combine both methodology, to reduce the spread of manuscript, which is anyway too long.”

Response 6.

Thank you for your valuable comment, which guided us to avoid the repetition of parts of the manuscript. Section "2.6.2. Rheology" has been changed as follows. Only points that differ from section "2.4.2.Rheology” are explained:

“The rheological properties of the optimized formulation were performed as de-scribed in the “2.4.2.Rheology” section. using a controlled stress rheometer (Haake Rheometer I Thermo Fisher Scientific Inc., Germany) with a parallel steel cone-plate geometry (35°TiL and 0.052 mm gap distance). However, the rheology studies of the optimized in situ gel were carried out at 3 temperatures, different from the section "2.4.2.Rheology". All analyses were performed at 5°C, 25°C, and 35°C to mimic storage, room, and ocular physiological temperature, respectively. iIn situ gel will be diluted with tear film after applied to the eye., For this reason, oscil-lation studies and Tsol/gel studies were also studied with in situ gel diluted with STF at 35°C different from the section "2.4.2.Rheology". Dilution was made in situ gel:STF(50:7) when the volume of a drop and the tear volume were calculated [32].”

Comment 7.

“What kind of Design of Experiment was carried out in the pre-formulation studies? How did the authors get 86 runs of experiment?”

Response 7.

Thank you for your comment. Table 3 belongs to the pre-formulation studies, and this part is shared for general information purposes only. The experimental design has not been applied.

Comment 8.

“Please improve the quality if Fig 1, 3 and 5, the fonts of axes are too small and blurred.”

Response 8.

We apologize for this problem. We increased the resolution of all figures in the manuscript to avoid any clarity issues. After this change, when we view the figures separately and in the manuscript, we do not experience any resolution problems. However, if you have any problems, please let us know.

Comment 9.

“The authors claim the optimized formulation contained 20% of Poloxamer 407 and 0.4% of poloxamer 188. Poloxamer 188 seems to have no effect in reaching the optimal sol-gel transition. What is the reason for applying it in the formulation?”

Response 9.

Thank you for your comment. Although Poloxamer 188 is used to a small extent in optimized formulation, as seen in Pareto charts in figure 1-a, it has a significant effect on Tsol/gel. In addition, this result is also compatible with the literature data we stated in lines 483-486.

Comment 10.

“The manuscript presents unnecessary data about other kind of polymers (Carbopol, HPMC) which showed no advantageous properties on the formulation. These unnecessary results could be omitted from manuscript, only in one sentence can be explained that they are not suitable for the purpose.”

Response 10.

Thank you for your comment. The reviewer's request made the pre-evaluation of the manuscript that the polymers used throughout the study, including the pre-formulation studies, should be evaluated for what purpose, positive or negative results in detail. For this reason, we have explained this section in this way. However, if you still find it appropriate to shorten it, we will make the necessary adjustments in consultation with the editors.

Comment 11.

“The discussion section is too long, pleas shorten it by focusing on the most important results of work.”

Response 11.

Thank you for your comment. As we stated in Comment 10, it is the request of the reviewer who made the pre-evaluation that the discussion section should be detailed to explain every detail. For this reason, we have explained this section in this way. However, if you still find it appropriate to shorten it, we will make the necessary adjustments in consultation with the editors.

Round 2

Reviewer 1 Report

Overall comment: The additions to the text clearly improved the quality of the manuscript. The language has been improved that is a good point. However, there are typos throughout the text (not obligatorily English related) that require the authors a careful proof reading to remove them.

Comment 1: The language has been improved that is a good point. However, there are typos throughout the text (not English related) that require the authors a careful proof reading to remove them. Below are some examples:

Line 54 : “impair” not “impare”

Line 119 : “0.45 Billerica µm”, in that example „Billerica“ has to be removed

Line 190 : “qbd” change for “QBD”

Line 202 : CMQ is employed and described nowhere

Comment 2:

Line 194 : Mixture design has been used as a general term, however a mixture design is often described a specific type of design of experiments (DoE). As Minitab was used for the regression you might find it if you look at Stat, DOE, Mixture. Please find another term, as it is not the methodology employed here.

Comment 3:

On the rebuttal of comment 3, the article was indeed not focused on algorithm development especially as the regression was performed with MINITAB. However, showing the modeling equations is paramount as the r2 is mentioned, at least add them in the supplementary data and refer to them in the text where the fit is described.

Author Response

General Comments: The additions to the text clearly improved the quality of the manuscript. The language has been improved that is a good point. However, there are typos throughout the text (not obligatorily English related) that require the authors a careful proof reading to remove them.

Response: Thank you for your comment and encouragement that our manuscript is promising. We are sorry about our typographical errors and grammatical mistakes. The language revision and edition of the text were conducted according to your warnings. If you have any problems related to language revision, please let us know.

Specific Comments:

Comment 1.

“The language has been improved that is a good point. However, there are typos throughout the text (not English related) that require the authors a careful proof reading to remove them. Below are some examples:

Line 54 : “impair” not “impare”

Line 119 : “0.45 Billerica µm”, in that example „Billerica“ has to be removed

Line 190 : “qbd” change for “QBD”

Line 202 : CMQ is employed and described nowhere”

Response 1.

Thank you for your comment and encouragement that our manuscript is promising. We are sorry about our typographical errors and grammatical mistakes. The language revision and edition of the text were conducted according to your warnings. If you have any problems related to language revision, please let us know.

Comment 2.

“Line 194 : Mixture design has been used as a general term, however a mixture design is often described a specific type of design of experiments (DoE). As Minitab was used for the regression you might find it if you look at Stat, DOE, Mixture. Please find another term, as it is not the methodology employed here.”

Response 2.

Thank you for your comments. The Minitab can also evaluate the design area not designed by the program with the custom design option. In the study, three-level full factorial and only one surfactant-containing study were evaluated with this option. In the article “the mixture design” was changed to “To see this interaction, a three-level full factorial design was applied, on the other hand, to see just one surfactant’s effects on the formulations, the formulations were produced containing only 1 surfactant.” (Lines 192-194)

Comment 3.

“On the rebuttal of comment 3, the article was indeed not focused on algorithm development especially as the regression was performed with MINITAB. However, showing the modeling equations is paramount as the r2 is mentioned, at least add them in the supplementary data and refer to them in the text where the fit is described.”

Response 3.

Thank you for your comments. We submit the equations as a supplementary data file. r2 value wee given in Table 9 at the manuscript. We also referred to the text supplementary data (Lines 372-373).

Reviewer 2 Report

The authors adressed all my concerns!

Author Response

Thank you for your comment and encouragement that our manuscript is promising.

This manuscript is a resubmission of an earlier submission. The following is a list of the peer review reports and author responses from that submission.

Round 1

Reviewer 1 Report

The paper presents the development of an in situ gelling ophthalmic preparation using QbD approach. The ophthalmic compositions contain posaconazole encapsulated in TPGS micelles, the ideal gel matrix for this micellar system being formulated in this article. Authors used macroscopic observations and rheological methods for optimization, after that further studies are performed only on the optimized composition.

Although the aim of the article is promising and interesting, the presentation and interpretation of the results are very incomplete and weak, the results and the related conclusion are not clearly.

Some serious critical remarks about the manuscript:

Why Carbopol has been studied as a temperature gelling system in the selection of the polymer matrix, and even alginates and xanthan gum are also mentioned from this aspect in the discussion.

The selected rheological parameters, except the gelation temperaturem cannot be interpreted, their descriptions are incomplete, the correlations derived from them are not clear (what is the meaning of thixotropy from the point of view of formulas, what is τ0 (yield stress?), how was it calculated what conclusions can be derived, how can this value be minus).

Gelling capacity measurement cannot be interpreted, what justified the study? (There is dilution with tear fluid later, so this test cannot be interpreted by the reader)

The texture analysis part cannot be interpreted, according to my opinion the measured adhesion is not related to the expected mucoadhesiveness, the other parameters are also not related to the efficiency of the optimized composition.

The in vitro release test is more likely to be considered an erosion, it is not clear from the method description whether the drug released from the micelles was actually measured.

The aim of this article is to formulate an optimal gel matrix for PSC-containing micelles, however, the article does not include any study to elucidate the relationship between drug micelles and gel matrix (not even blank matrices have been studied).

The Qbd approach is incomplete. QbD tools: CQAs are not justified, the fitted model is not detailed.